# Development of folate receptor targeting chimeras for cancer selective degradation of extracellular proteins

Yaxian Zhou[1], Chunrong Li[1], Xuankun Chen [1], Yuan Zhao[1], Yaxian Liao[2], Penghsuan Huang [2], Wenxin Wu[2], Nicholas S. Nieto[1], Lingjun Li [1,2] & Weiping Tang [1,2] ✉

Targeted protein degradation has emerged as a novel therapeutic modality to treat human diseases by utilizing the cell's own disposal systems to remove protein target. Significant clinical benefits have been observed for degrading many intracellular proteins. Recently, the degradation of extracellular proteins in the lysosome has been developed. However, there have been limited successes in selectively degrading protein targets in disease-relevant cells or tissues, which would greatly enhance the development of precision medicine. Additionally, most degraders are not readily available due to their complexity. We report a class of easily accessible Folate Receptor TArgeting Chimeras (FRTACs) to recruit the folate receptor, primarily expressed on malignant cells, to degrade extracellular soluble and membrane cancer-related proteins in vitro and in vivo. Our results indicate that FRTAC is a general platform for developing more precise and effective chemical probes and therapeutics for the study and treatment of cancers.

Targeted protein degradation (TPD) exploiting the cells' own degradation machinery to remove the protein of interest (POI) is emerging as a novel therapeutic modality[1,2]. There are numerous advantages of TPD over traditional strategies that block the functional sites, such as more sustained response and potential accessibility to the undruggable targets[3–5]. PROteolysis TArgeting Chimeras (PROTACs) were developed first and received the most attention[6–9]. Over a dozen PROTACs that recruit cereblon or Von Hippel-Lindau E3 ligases have progressed into human clinical trials for the treatment of cancer and other diseases[10]. In addition, about one dozen other E3 ligases were demonstrated in their utility for the development of PROTACs[11], such as RNF114[12] and DCAF16[13]. Although many E3 ubiquitin ligases have different expression profiles in tissues[14], they have not been exploited for selective degradation of proteins in the disease-relevant cells or tissues due to the lack of selective cell permeable ligands. In addition to PROTACs, many platforms have been developed to degrade the intracellular POI through other mechanisms[15–17]. In contrast, fewer strategies are available for the degradation of extracellular proteins[18], though they are consisted of 40% of the proteome[19].

The initial LYsosome TArgeting Chimeras (LYTACs), composed of a polymeric glycopeptide ligand of cation-independent mannose-6-phosphate receptor (CIM6PR) and a binder of the extracellular POI, were reported in 2020[20]. LYTACs could induce the degradation of a broad range of soluble and membrane protein targets in many tissues due to the ubiquitous expression profile of CIM6PR. Later, several groups including ours reported LYTACs with tissue selectivity by recruiting asialoglycoprotein receptor (ASGPR) to restrict the TPD in liver cells only[21–23]. Before we started our work, only lectins had been employed as the lysosomal targeting receptor (LTR) for TPD. While we were preparing this manuscript, degraders that recruit non-lectin LTRs were reported for the degradation of extracellular proteins[18,24,25]. Integrin and transferrin receptor 1, overexpressed in cancer cells, have been recently leveraged for TPD. Degraders that recruit these receptors hold the possibility of selectively depleting the extracellular

[1]Lachman Institute of Pharmaceutical Development, School of Pharmacy, University of Wisconsin-Madison, Madison, WI 53705, USA. [2]Department of Chemistry, University of Wisconsin-Madison, Madison, WI 53706, USA. ✉e-mail: weiping.tang@wisc.edu

proteins in cancer cells. However, the cancer selectivity has not been examined[25,26].

Similar to CIM6PR and ASGPR, the folate receptor (FR) has also been investigated for drug delivery[27,28]. FR has been recognized as a major biomarker for tumor cells. The overexpression of FR is found in various cancer cells, such as ovarian cancer, non-small-cell lung cancer (NSCLC), and myeloid leukemia, while most normal tissues lack the expression of FR on the cell surface[29–31]. We reasoned that it would be ideal to recruit FR to selectively degrade extracellular proteins associated with cancers in cancer cells to achieve high efficiency and selectivity.

Two isoforms of FR, FR1 and FR2, are glycosylphosphatidylinositol-anchored membrane proteins functioning as high-affinity receptors for folate (FA) to import FA into the cell via receptor-mediated endocytosis[32,33]. The detectable FR expression on normal tissues, predominantly in the kidney, lung, and choroid plexus, is largely restricted to the apical surface of the polarized epithelial cells, preventing the exposure of FR to the folate-drug conjugates in the blood circulation, thus reducing cytotoxic effects on healthy tissues. Upon tumorigenesis, the change in tissue architecture enables FR to be accessible to the circulating folate-drug conjugates[34–36]. These unique expression features make FR a promising receptor for cancer-selective targeting. FA, the ligand of FR, has the advantages of non-immunogenic, low cost, high stability, and maintaining high binding affinity to FR after conjugation[37,38]. It has been reported that the folate-conjugate is engulfed by the cell through FR-mediated endocytosis followed by the transportation into endosome and lysosome. The drop in pH can trigger the release of folate-conjugate into the cell, while the FR is recycled back to the membrane for the transport of more folate-conjugates[39–41]. Currently, various folate-tethered drugs in the forms of folate-nanoparticles, small-molecule drug conjugates, and radio-immunoconjugates are being investigated for delivering imaging and therapeutic reagents into tumors[42]. While most of these reagents are still undergoing clinical trials, a FA-based fluorescent imaging reagent (Cytalux)[43] and an ADC targeting FR (Elahere)[44] were approved for cancer surgery and treatment, respectively. Recently, the FR-targeting strategy has been explored in the field of TPD. Studies have demonstrated that attaching folate to molecular glues or PROTACs allows for selective degrading cytosolic proteins in cancer cells[45,46]. Herein, we report our development of Folate Receptor TArgeting Chimeras (FRTACs) as a general platform for selectively degrading extracellular cancer-relevant proteins in cancer cells (Fig.1a). The FRTACs are composed of a FA ligand that can bind to FR and antibodies that can bind to cancer-relevant targets. We demonstrate that the FRTACs are able to mediate the degradation of protein targets in vitro and in vivo. The FRTAC shows more significant tumor growth suppression than the corresponding blocking antibody in three different syngeneic mouse cancer models. Most importantly, the FRTACs can be easily obtained from commercially available FA conjugating reagents and antibodies against any extracellular target. Our results support that FRTACs can be a precise and effective technology for the study and treatment of cancers.

## Results

### Degradation of soluble proteins mediated by FRTACs in vitro

Antibody-based FRTAC was generated through a two-step labeling method by treating the antibody sequentially with two commercially available reagents, cyclic alkyne (DBCO-PEG3-NHS ester) and azido-PEG-linker (MW = 2 K)-folate, which could be coupled together through click chemistry (Fig.1b). To test whether the soluble protein can be taken into cells by FRTACs, we used goat anti-mouse IgG antibody attached with folate (Ab-FA) and folate-PEG (MW = 2 K)-FITC (FA-FITC) as antibody- and small molecule-based FRTACs, respectively, to mediate the endocytosis of a model target - mouse anti-FITC-594 (Fig. 1c). HepG2 cells were treated with 50 nM anti-FITC-594 and increasing concentrations of each degrader for 3 h. The results showed

that Ab-FA induced the cellular uptake of anti-FITC-594 in a dose-dependent manner, while the uptake of anti-FITC-594 mediated by FA-FITC peaked at 200 nM and decreased at 1000 nM, reflecting a typical hook effect of small molecule degraders, where more FA-FITC/anti-FITC-594 or FA-FITC/FR binary complexes are formed than the ternary complex of FR/FA-FITC/anti-FITC-594 (Supplementary Fig. 1a,b). By directly comparing the cellular uptake of anti-FITC-594 mediated by 50 nM Ab-FA and 200 nM FA-FITC, a significantly greater amount of internalized anti-FITC-594 was observed in the cells treated with Ab-FA than FA-FITC, suggesting that antibody-based degrader Ab-FA has higher uptake efficiency over FA-FITC (Fig. 1d). In addition to anti-FITC-594, the uptake of mouse anti-biotin (−594 or −647) mediated by Ab-FA further verified that different soluble proteins can be internalized by FA-conjugates. Antibody without folate modification cannot take the anti-biotin-594 into the cells (Supplementary Fig. 1c).

We then monitored the degradation of internalized protein target by a pulse-and-chase assay. HepG2 cells were first treated with Ab-FA and anti-biotin-647 for 3 h to allow the cellular uptake of target, followed by another 3 h incubation in fresh media with or without lysosomal degradation inhibitors Chloroquine (CQ), Bafilomycin A1 (BAF), or the proteasome degradation inhibitor MG132. We found that the level of anti-biotin-647 accumulated in the cells within the first 3 h was significantly decreased after the removal of the degrader and target from the media, suggesting the degradation of target occurred after internalization. The treatment of CQ and BAF significantly inhibited the degradation of anti-biotin-647, while MG132 did not rescue the reduction of anti-biotin-647, which indicated that the FRTAC routes the protein target into lysosome rather than proteasome for degradation (Fig. 1e). To further demonstrate that the internalized soluble protein was delivered into lysosome for degradation, we treated Hela cells with 25 nM Ab, 25 nM Ab plus 125 nM free FA, or 25 nM Ab-FA together with 50 nM anti-Rabbit-647 for 24 h, and found that Ab-FA significantly increased the intracellular level of anti-Rabbit-647 compared to the other two groups. By staining the live cells with Lyso-Tracker, we also observed the colocalization of internalized anti-Rabbit-647 with lysosomes in the presence of Ab-FA (Fig. 1f). Rab7 is an essential protein that mediates the late endosome/lysosome trafficking[47]. We then downregulated Rab7 level in Hela cells by siRNA and compared the amount of anti-Rabbit-647 in the cells transfected with scramble or Rab7 siRNA. The results showed that the level of anti-Rabbit-647 was higher in the Rab7 knockdown cells compared to the control when co-treated with Ab-FA, suggesting that anti-Rabbit-647 was accumulated in the cells due to reduced degradation, resulting from the disrupted transport from late endosome to lysosome caused by Rab7 knockdown (Fig. 1g). Overall, all data support that FRTAC delivers extracellular soluble proteins into lysosome for degradation.

Next, we treated cells with the bifunctional degrader and excess free folate, which can occupy FR to prevent its interaction with the degrader. The results showed that the uptake of anti-biotin-647 in HepG2 cells was significantly decreased in the presence of excess free folate, which indicates the involvement of FR in the process of protein target internalization (Fig. 2a). To further validate the role of FR in transporting FRTAC/target protein into the cells, we downregulated the expression of FR1, the dominant isotype of FR, in Hela cells by siRNA. After incubating with Ab-FA for 6 h, a significantly lower amount of internalized anti-Rabbit-647 was detected in the FR1 knockdown cells than the cells transfected with scrambled siRNA (Fig. 2b). Additionally, we treated Hela cells that transiently over-expressing FR1-FLAG with Ab-FA and anti-Rabbit-647 for 6 h, and the results showed that compared to the non-transfected cells and cells transfected with empty vector, the elevation of FR1 expression level significantly boosted the uptake of anti-Rabbit-647 mediated by Ab-FA (Fig. 2c). Interestingly, we observed that Hela cells overexpressing FR2 also exhibited enhanced anti-Rabbit-647 internalized when co-treated with Ab-FA, suggesting that FRTAC can interact with both isotypes of

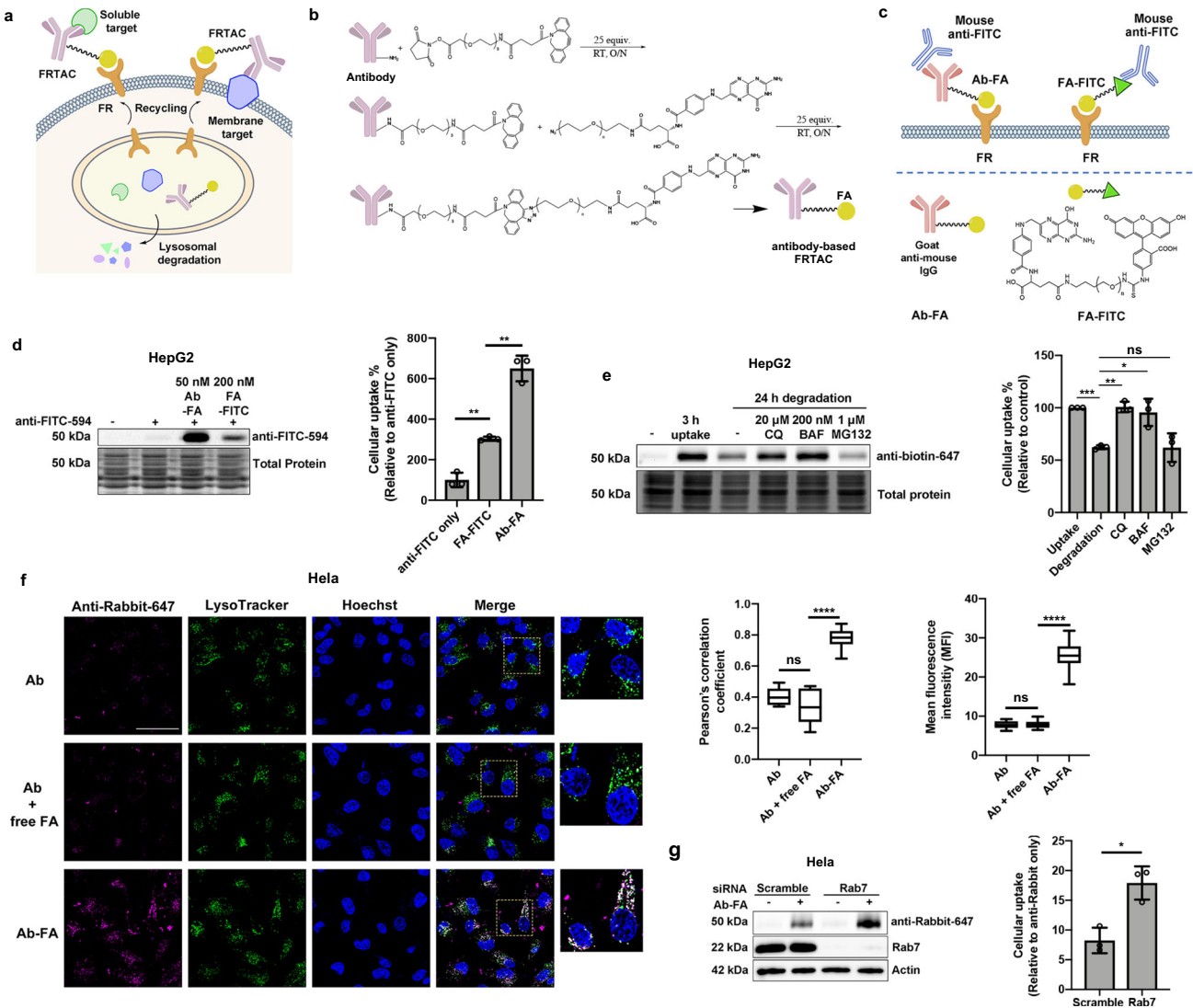

**Fig. 1 | FRTACs mediate the uptake and lysosomal degradation of soluble proteins. a** Illustration of FRTAC-induced lysosomal targeted protein degradation. **b** Generation of FRTAC through a two-step labeling method. **c** Schematic of mouse anti-FITC uptake mediated by Ab-FA and FA-FITC. **d** Uptake of anti-FITC-594 (50 nM) in HepG2 cells treated with Ab-FA (25 nM) and FA-FITC (200 nM) for 3 h (n = 3). **e** In-gel fluorescence analysis of anti-biotin-647 (50 nM) internalization and degradation in HepG2 cells by Ab-FA (25 nM) in the presence or absence of Chloroquine (CQ, 20 μM), Bafilomycin A1 (BAF, 200 nM), and MG132 (1 μM) for 24 h (n = 3). **f** Cellular uptake and lysosome colocalization of anti-Rabbit-647 (50 nM) in the presence of Ab (25 nM), Ab (25 nM) + free FA (125 nM), and Ab-FA (25 nM) in Hela cells for 24 h by immunofluorescent staining. Scale bar: 50 μm. The colocalization of internalized anti-Rabbit-647 with lysosomes was analyzed by Pearson's correlation coefficients. The intracellular fluorescence intensity is presented as mean fluorescence intensity (MFI) (n = 15 images from three biologically independent experiments). Box plot: minima (lower whisker), maxima (upper whisker), center (median), bounds of the box (25th and 75th percentiles), whiskers (range from minima to maxima). **g.** Uptake of anti-Rabbit-647 (50 nM) mediated by Ab-FA (25 nM) in Hela cells transfected with scramble siRNA or Rab7 siRNA for 3 h (n = 3). N indicates biologically independent experiments except for Fig. 2f. Data are presented as mean ± SD. The statistical significance was assessed using an unpaired two-tailed t test, *P < 0.05, **P < 0.01, ***P < 0.001, ****P < 0.0001, ns: not significant. Source data are provided as a Source Data file.

FR to mediate the target protein delivery (Fig. 2d). To elucidate the endocytosis pathway utilized by FRTAC to transport soluble proteins into the cells, we pre-treated cells with methyl-β-cyclodextrin (MβCD, inhibitor of caveolae/lipid-raft-mediated endocytosis), chlorpromazine (CHP, inhibitor of clathrin-mediated endocytosis) or cytochalasin D (cyto-D, inhibitor of macropinocytosis) for 1 h and then incubated with 25 nM Ab-FA and 50 nM anti-FITC-594 for 3 h. The results showed that the treatment of MβCD significantly blocked the cellular uptake of anti-FITC-594 induced by Ab-FA, while CHP and cyto-D failed to reduce the level of internalized anti-FITC-594, which indicated that Ab-FA mediated the soluble protein internalization primarily through caveolae/lipid-raft-mediated endocytosis, which is consistent with the endocytosis pathway reported for FR (Fig. 2e)[47–49]. Last, we compared the uptake of anti-biotin-594 mediated by Ab-FA across different cancer cell lines (Fig. 2f and Supplementary Fig. 1d). The expression levels of FRs on these cell lines were quantified with FA-FITC by flow cytometry (Fig.2g, Supplementary Fig. 9 and Supplementary Table 2). The results showed that the uptake efficiency of soluble proteins varied among different cancer cell lines and well correlated with corresponding folate receptor level (Fig. 2h). All of the above data support the involvement of FR in the endocytosis process.

## Degradation of endogenous membrane proteins mediated by FRTACs in vitro

After demonstrating that FRTACs can induce the uptake and degradation of model soluble proteins via their interaction with FR, we next evaluated the possibility of selectively degrading membrane proteins in cancer cells. We previously attached tri-GalNAc-NHS ester to

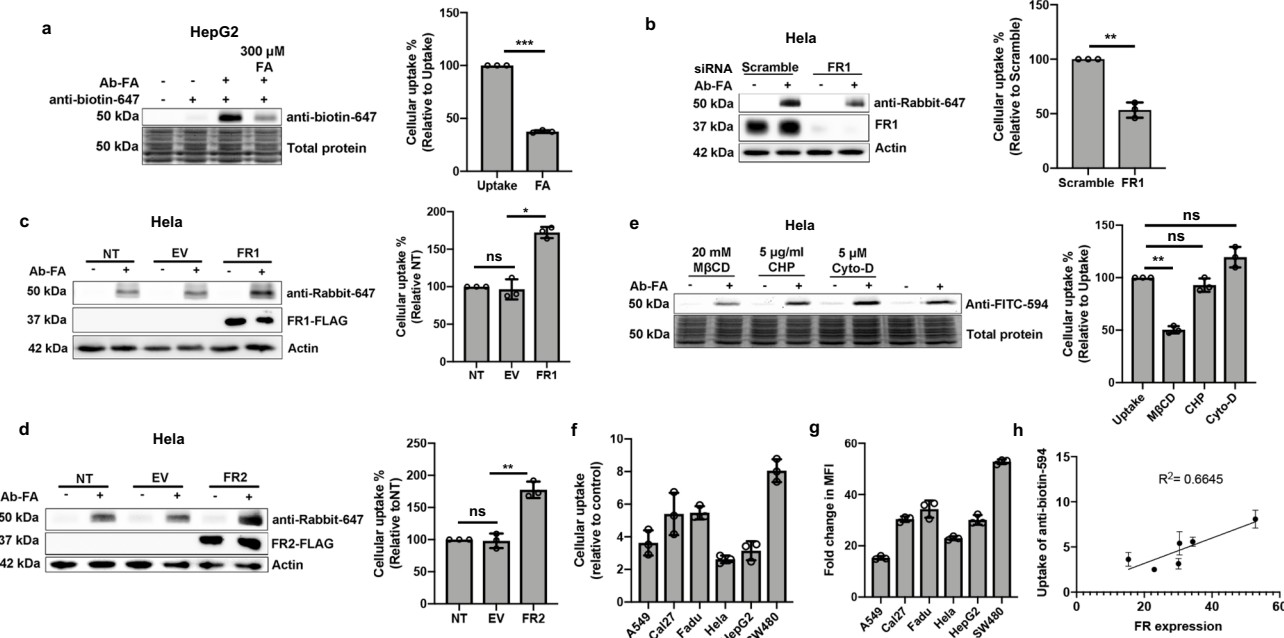

**Fig. 2 | FRTACs recruit FR to induce lysosomal degradation of soluble proteins.**
**a** Inhibition of anti-biotin-647 (50 nM) internalization in the presence of Ab-FA (25 nM) by free FA (300 µM) in HepG2 cells for 3 h (n = 3). **b** Uptake of anti-rabbit-647 (50 nM) mediated by Ab-FA (50 nM) in Hela cells transfected with scramble or FR1 siRNA for 6 h (n = 3). **c** Uptake of anti-rabbit-647 (50 nM) mediated by Ab-FA (50 nM) in Hela cells transfected with plasmid expressing FLAG-FR1 for 6 h. Non-transfected (NT) cells and cells transfected with empty vector (EV) were used as negative controls (n = 3). **d** Uptake of anti-Rabbit-647 (50 nM) in FR2 over-expression cells. Non-transfected (NT) cells and cells transfected with empty vector (EV) were used as negative controls (n = 3). **e** Inhibition of anti-FITC-594 (50 nM)

internalization in the presence of Ab-FA (25 nM) by methyl-β-cyclodextrin (MβCD, 20 mM), chlorpromazine (CHP, 5 µg/ml) or cytochalasin D (cyto-D, 5 µM) in Hela cells for 3 h (n = 3). **f** Uptake of anti-biotin-594 (50 nM) in different cancer cell lines (n = 3). **g** Quantification of FR expression levels on different cancer cell lines by flow cytometry (n = 3). **h** Correlation of uptake efficiency with FR expression levels on different cancer cell lines (n = 3). N indicates biologically independent experiments. Data are presented as mean ± SD. The statistical significance was assessed using an unpaired two-tailed t test, *P < 0.05, **P < 0.01, ***P < 0.001, ns: not significant. Source data are provided as a Source Data file.

Cetuximab (Ctx), the therapeutic antibody against epidermal growth factor receptor (EGFR), by reacting with lysine residues on the antibody[22]. These tri-GalNAc conjugates can recruit ASGPR and mediate the degradation of EGFR selectively in liver cells. Using the same method, Ctx was reacted with 3, 12, or 25 equivalent of commercially available folate-NHS ester to generate degraders with increasing numbers of ligands on each antibody (Supplementary Fig. 2a). Our data indicated that a higher degree of folate labeling led to greater EGFR degradation at both 10 nM and 200 nM concentrations (Supplementary Fig. 2b). To further improve the degradation efficiency of FRTAC on membrane targets, we compared this direct labeling method using folate-NHS ester with the two-step labeling method as mentioned above. We tried two different PEG-linkers and controlled the degree of FA labeling on each antibody by adjusting the equivalence of DBCO-PEG3-NHS ester in the reaction. Our results showed that consistent with earlier findings, degraders produced by both methods exhibited higher EGFR degradation efficiency when labeled with more folate. Degraders bearing either a 1k-PEG-linker or a 2k-PEG-linker reduced EGFR level to the similar extent when the degraders are generated via the same labeling method. While most degraders with the same linker length and degree of labeling showed no significant difference in EGFR degradation efficiency between the one-step and two-step labeling methods, the degrader with a 2k-PEG-linker from the two-step labeling method (N3, 2k, 25x) showed higher degradation efficiency than the degrader with either 1k or 2k PEG-linker from one-step labeling method (Supplementary Fig. 2c).

The FRTAC (Ctx-FA) with a 2k-PEG-linker introduced by click chemistry was selected for the following studies due to more

efficient EGFR degradation detected (Fig. 3a). First, the dose- and time-dependency of the EGFR degradation mediated by Ctx-FA was verified in Fadu and Hela cells. Ctx-FA displayed a DC$_{50}$ of 0.41 nM and a D$_{max}$ of 75% in Fadu cells, while it induced the EGFR degradation with a DC$_{50}$ of 0.24 nM and a D$_{max}$ of 80% in Hela cells (Fig. 3b and Supplementary Fig. 3a). The EGFR degradation became significant around 4–6 h post-treatment and the maximal degradation lasted at least till 72 h (Fig. 3c and Supplementary Fig. 3b). No EGFR degradation was detected in Fadu cells treated with Ctx, free FA, the combination of Ctx and free FA, or FA attached human IgG isotype (Fig. 3d and Supplementary Fig. 3c). The degradation induced by Ctx-FA was confirmed by confocal imaging in Fadu cells, which revealed not only a significant decrease of EGFR in the cells treated with the degrader compared to control groups, but also a clear translocation of EGFR from the cell surface into the cytoplasm after 24 h of Ctx-FA treatment (Fig. 3d). To track the EGFR after internalization, we co-stained EGFR and lysosome marker LAMP1 after incubating with the degrader and found that EGFR was primarily co-localized with lysosomes (Fig. 3d). Our results also showed that the degradation of EGFR was partially abolished when the cells were treated with the lysosomal degradation inhibitor, Bafilomycin A1, while the proteasome inhibitor MG132 was not able to rescue EGFR degradation in the presence of Ctx-FA, demonstrating that Ctx-FA triggered EGFR degradation involves lysosome but not proteasome (Fig. 3e, f). Moreover, we found that decreasing the level of Rab7 partially prohibited the depletion of EGFR mediated by Ctx-FA, further confirming that FRTAC could transport membrane protein into lysosome for degradation (Fig. 3g).

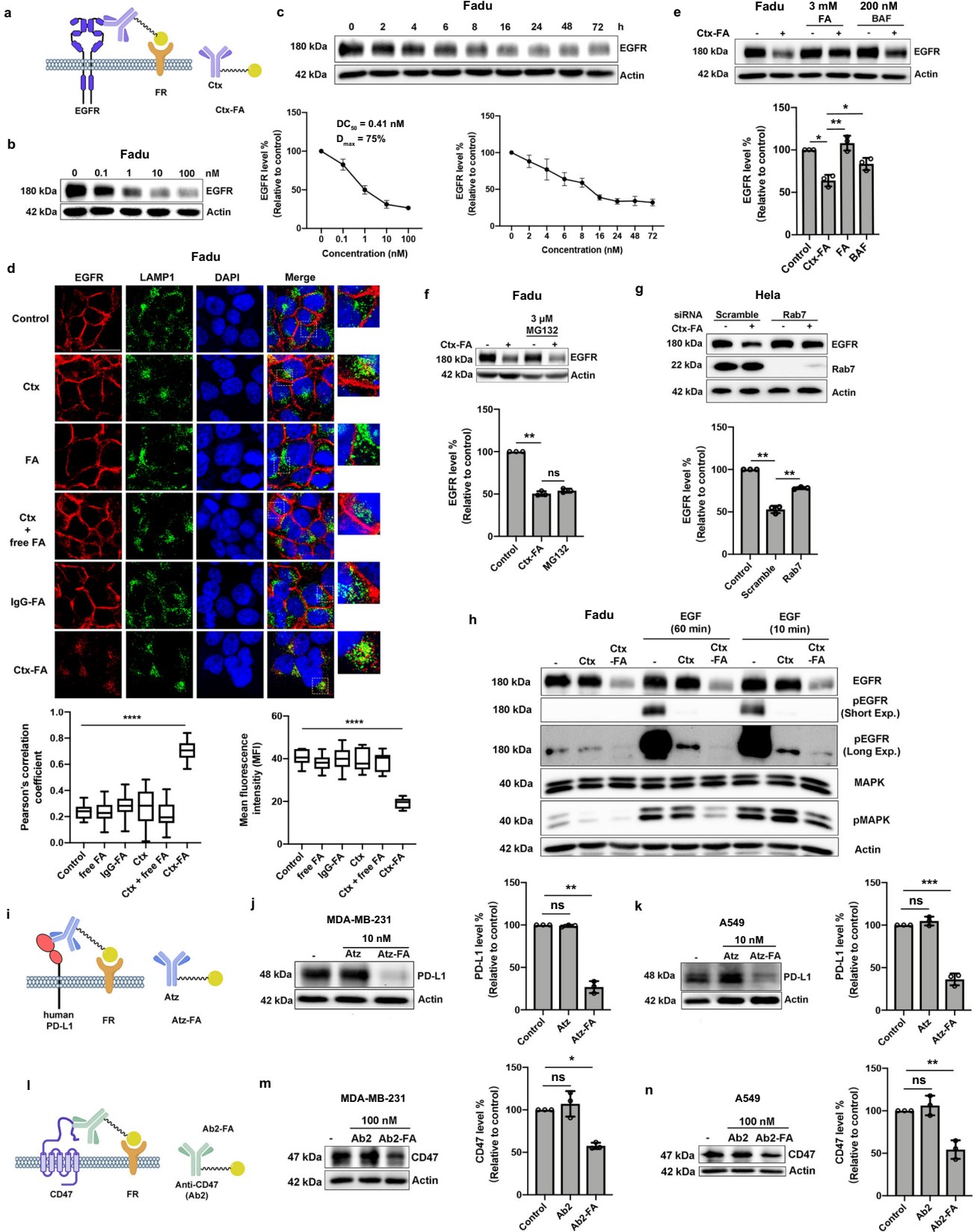

**Fig. 3 | FRTACs mediate the lysosomal degradation of membrane proteins (EGFR, PD-L1, and CD47) via their interaction with FR. a** Schematic of Ctx-FA targeting EGFR. **b** Dose response of EGFR degradation (24 h) in Fadu cells (n = 3). **c** Time course of EGFR degradation mediated by Ctx-FA (10 nM) in Fadu cells (n = 3). **d** Immunofluorescent staining of EGFR degradation and lysosome colocalization after treatment of Ctx (10 nM), FA (50 nM), Ctx (10 nM) + free FA (50 nM), IgG-FA (10 nM) and Ctx-FA (10 nM) for 24 h in Fadu cells. Scale bar: 25 μm. The colocalization was analyzed by Pearson's correlation coefficients (n = 15). The intracellular fluorescence intensity is presented as mean fluorescence intensity (MFI) (n = 10). Box plot: minima (lower whisker), maxima (upper whisker), center (median), bounds of the box (25th and 75th percentiles), whiskers (range from minima to maxima). **e** Inhibition of EGFR degradation in the presence of Ctx-FA (10 nM) by free FA (3 mM) and Bafilomycin A1 (BAF, 200 nM) in Fadu cells for 6 h (n = 3).

**f** Inhibition of EGFR degradation in the presence of Ctx-FA (10 nM) by MG132 (3 μM) in Fadu cells for 6 h (n = 3). **g** EGFR degradation mediated by Ctx-FA (10 nM) in Hela cells transfected with scramble siRNA or Rab7 siRNA for 6 h (n = 3). **h** Downregulation of EGFR and MAPK phosphorylation in Fadu cells. Representative blots from three biologically independent experiments. **i**. Schematic of Atz-FA targeting PD-L1. **j**, **k** Degradation of PD-L1 in MDA-MB-231 and A549 cells treated with Atz-FA (10 nM) for 24 h (n = 3). **l** Schematic of Ab2-FA targeting CD47. **m**, **n** Degradation of CD47 in MDA-MB-231 and A549 cells treated with Ab2-FA (100 nM) for 24 h (n = 3). N indicates images from three biologically independent experiments (**d**) or biologically independent experiments (**b**, **c**, **e-g**, **j**, **k**, **m**, **n**). Data are presented as mean ± SD. The statistical significance was assessed using an unpaired two-tailed t test, *P < 0.05, **P < 0.01, ***P < 0.001, ****P < 0.0001, ns: not significant. Source data are provided as a Source Data file.

To investigate the involvement of FR in degrader-induced EGFR degradation, Fadu cells were incubated with excess free folate and the results showed that less EGFR was degraded in the presence of excess free folate, suggesting the engagement of FR in inducing EGFR degradation (Fig. 3e). No significant difference in the FR1 and FR2 levels was detected after the degrader treatment, indicating that FRTAC only routes the protein target for degradation without affecting the recycling of the FRs (Supplementary Fig. 3d). We also determined EGFR degradation in various cancer cell lines and analyzed their correlation with FR expression levels on each cell line. Our results suggested that unlike the uptake of soluble protein, the EGFR degradation efficiency of the degrader was related not only to the expression level of FR but also to the expression level of EGFR in different cell lines. The degradation efficiency of Ctx-FA correlated more strongly with the ratio of FR to EGFR expression levels than with FR expression level alone, due to the involvement of both proteins in the ternary complex. A higher FR to EGFR expression ratio resulted in increased EGFR degradation efficiency (Supplementary Fig. 3e–j). Overall, these data demonstrate that the lysosomal degradation of EGFR induced by the degrader is mediated through the interaction between folate and FR.

Lastly, we investigated how Ctx-FA regulates the EGFR downstream signaling pathway by monitoring the phosphorylation of EGFR and MAPK upon activation. We found that the treatment of Ctx-FA downregulated the level of EGFR and abolished the phosphorylation of EGFR compared to the untreated cells after the stimulation with 100 ng/mL EGF for either 10 or 60 min. The level of p-MAPK, though increased after EGF stimulation, was suppressed after Ctx-FA treatment compared to the antibody-treated and untreated cells, while the expression of MAPK was not changed in the presence of Ctx-FA (Fig. 3h). Our results indicate that the FRTAC can downregulate EGFR level in cancer cells and further modulate the associated downstream signaling pathway.

In addition to EGFR, we explored whether FRTACs can be applied to two other therapeutic targets, PD-L1 and CD47, which are both highly expressed on cancer cells. Atezolizumab (Atz), the therapeutic antibody against PD-L1, and anti-CD47 monoclonal antibody were reacted with DBCO-NHS ester and folate-azide sequentially to form the degraders against PD-L1 (Atz-FA) and CD47 (Ab2-FA), respectively. The treatment of MDA-MB-231 and A549 cells with 10 nM Atz-FA led to significant degradation of PD-L1 in both cell lines after 24 h (Fig. 3i–k). Similarly, the amount of CD47 was significantly reduced in MDA-MB-231 and A549 cells when incubated with 100 nM of degrader (Fig. 3l–n). These data indicate that FRTACs have a broad scope for the degradation of cancer-associated protein targets.

### Anti-tumor effect of FRTACs targeting PD-L1 in vivo

Encouraged by the in vitro degradation of several membrane protein targets in multiple cancer cell lines, we then investigated the effect of the FRTACs in vivo. PD-L1 is an immune checkpoint interacting with its receptor PD-1 on the immune cells to induce T cell immunosuppression. Its role in assisting the evasion of cancer cells from host

immune surveillance makes PD-L1 a validated target for cancer immunotherapy[50,51]. However, current therapeutics targeting PD-L1 has low patient response rate and significant side effects due to undesired inhibition of PD-L1 on normal tissues[52–54]. We hypothesize that degraders specifically targeting PD-L1 in cancer cells may have the potential to be a more selective therapeutic with higher efficacy and better safety profiles. To test this hypothesis, we first generated the FRTAC targeting mouse PD-L1 (Ab3-FA) using rat anti-mouse PD-L1 antibody (Ab3) through the two-step labeling method (Supplementary Fig. 4a). The binding affinity of Ab3-FA with mouse PD-L1 was determined as 24.6 ± 4.67 nM using Micro-Scale Thermophoresis (MST) assay, which is comparable to the affinity between unmodified antibody and mouse PD-L1 (Kd= 9.05 ± 0.78 nM), indicating the attachment of FA on the antibody does not significantly interfere the interaction between the FRTAC and target protein (Supplementary Fig. 4b). The degradation of mouse PD-L1 was then tested in murine colon cell line CT26 and melanoma cell line B16F10. Both cell lines were treated with 100 ng/mL IFNγ for 24 h to induce the PD-L1 expression before the treatment of anti-mouse PD-L1 antibody or degrader. We found that antibody slightly reduced mouse PD-L1 level in both CT26 and B16F10 cells, while FRTAC could result in more significant downregulation of PD-L1 in both cell lines compared to control and antibody-treated groups (CT26: $DC_{50} = 0.29$ nM, $D_{max} = 51\%$; B16F10: $DC_{50} = 0.52$ nM, $D_{max} = 64\%$; Supplementary Fig. 4c, d). Degradation occurred after 8 h post-treatment and lasted at least till 48 h (Supplementary Fig. 4e, f). Similar to EGFR, the degrader transported mouse PD-L1 into the lysosomal degradation pathway after interacting with FR on both cell lines (Supplementary Fig. 4g, h). In addition, we also demonstrated the degradation of PD-L1 in mouse head & neck cancer cell line MOC1 (Supplementary Fig. 4i).

We then evaluated the pharmacokinetics (PK) of the mouse PD-L1 degraders in vivo. To study whether the number of folate labeled on each antibody could affect the clearance rate of the degrader, degraders with 2–3 FA per antibody (Ab3-FA-12x) or 4-5 FA per antibody (Ab3-FA-25x) were generated by adjusting the equivalence of folate-azide in the second step of labeling. The PK of unmodified PD-L1 antibody and two PD-L1 degraders were determined by intraperitoneal (IP) injection at a dose of 2.5 mg/kg into C57BL/6 mice and blotting the Rat IgG level in the plasma at different time points. The analysis of PK in mice indicated that the $T_{1/2}$s for Ab3, Ab3-FA-12x, and Ab3-FA-25x were around 24 h, 18 h, and 15 h, respectively (Fig. 4a, and Supplementary Fig. 5a). The amount of FA ligands on the antibody has some effect on the PK, but the difference between Ab3-FA-12x and Ab3-FA-25x is relatively small. Because the latter has higher degradation efficiency in the cell-based assay, it was chosen for the following studies. Before testing the anti-tumor efficacy of the degrader in vivo, we evaluated the plasma concentration of Ab3-FA-25x in B16F10 tumor-bearing C57BL/6 mice by administering at a dose of 2.5 mg/kg via IP injection. The concentration peaked around 10 μg/ml at 3 h post-treatment, followed by a gradual decline to 5 μg/ml at 8 h and approached 0 μg/mL by 48 h (Fig. 4b, and Supplementary Fig. 5b).

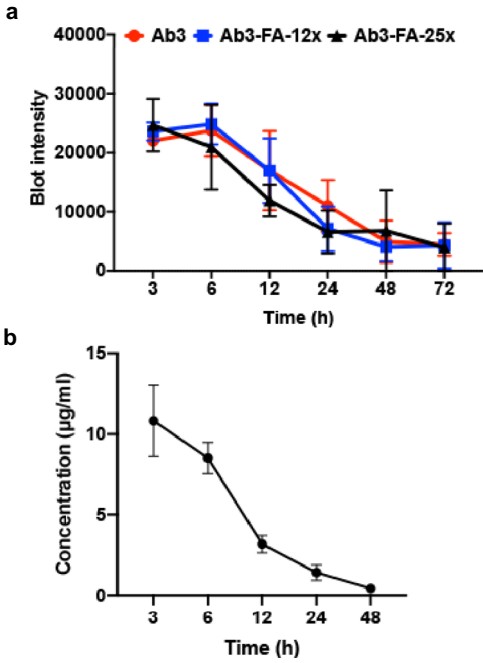

**Fig. 4 | Evaluation of the pharmacokinetics (PK) for PD-L1 degraders in vivo.**
**a** Blot intensity of rat IgG in the plasma of C57BL/6 mice treated with Ab3, Ab3-FA-12x, Ab3-FA-25x at different time points (2.5 mg/kg via IP injection). Data are presented as mean ± SD, n = 4. **b** Concentration of Ab3-FA-25x in C57BL/6 mice bearing B16F10 tumor at different time points (2.5 mg/kg via IP injection) Data are presented as mean ± SD, n = 3. (12x and 25x: 12 or 25 molar equivalents of DBCO-NHS ester in the first step; 25 equivalents of folate-azide were used in the second step). N indicates mice. Source data are provided as a Source Data file.

Next, we investigated whether our PD-L1 degrader can impede tumor growth in vivo. To assess the efficacy of the PD-L1 degrader across different types of cancer, we tested the anti-tumor effect in three different syngeneic mouse models. B16F10 tumor-bearing C57BL/6 mice, CT26 tumor-bearing BALB/c mice and MOC1 tumor-bearing C57BL/6 mice were randomized into three groups respectively. Mice in each group were treated with PBS, PD-L1 antibody (Ab3) or PD-L1 degrader (Ab3-FA) via IP injection, and continuously treated every other day for a total of 5 doses. 7.5 mg/kg/dose of Ab3 and Ab3-FA was administered into CT26 mouse model, while 2.5 mg/kg/dose was administered into B16F10 and MOC1 mouse (Fig. 5a–c). By monitoring the tumor size in-situ, we found that Ab3-FA could suppress tumor burden in all three mouse models. The most dramatic inhibitory effect was observed for MOC1 tumors, while CT26 and B16F10 tumor models showed less reduction on the tumor growth. Even though the mice exhibited varying response to the treatment, our PD-L1 degrader Ab3-FA consistently induced a more significant tumor growth delay compared to the Ab3-treated group in all three tumor models (Fig. 5d–f). No obvious adverse effect was observed throughout our studies. The body weight was similar among all three groups in all mouse models (Fig. 5g–i). We then isolated and analyzed the CT26 and MOC1 tumors at the end of the experiment. Our results showed that mice injected with the degrader had significant smaller tumor sizes and lower tumor weights than the antibody-treated mice (Fig. 5j, k). In addition, we treated 4 groups of CT26 tumor-bearing mice with PBS, IgG-FA (7.5 mg/kg), Ab3 (7.5 mg/kg) and the combination of Ab3 (7.5 mg/kg) and free FA (0.11 mg/kg) respectively via IP injection every other day for a total of 5 doses. We found that the non-targeting FA-conjugated IgG did not reduce the tumor growth compared to PBS-treated group. Mice treated with Ab3 or Ab3 + free FA exhibited similar trends in tumor growth, indicating that the addition of free FA had no effect on tumor progression (Supplementary Fig. 6). These data

suggest that our PD-L1 degrader could effectively slow the tumor progression in multiple tumor models, and exhibit more potent anti-tumor efficacy than the blocking antibody.

We then tried to elucidate the potential mechanism underlying the anti-tumor effect of Ab3-FA in CT26 mouse model. Mice bearing 100–200 mm³ CT26 tumors were randomly divided into five groups (n = 3), and received PBS, 7.5 mg/kg IgG-FA, the combination of 7.5 mg/kg Ab3 plus 0.11 mg/kg free FA, 7.5 mg/kg Ab3, or 7.5 mg/kg Ab3-FA respectively via IP injection continuously for 3 days. Tumors were collected on day 4 and analyzed for protein levels of PD-L1 and CD8 (a marker for cytotoxic T cell population) by western blot and immunohistochemistry (IHC) staining. We found that Ab3-FA induced a remarkable decrease in tumorous PD-L1 and an increase in CD8 in the tumor tissues compared to the control groups. The FA conjugated isotype and the addition of free FA did not reduce PD-L1 level nor enhance CD8 compared to the mice treated with PBS or Ab3, respectively, indicating FA alone cannot influence the target protein level and CD8+ T cell infiltration (Supplementary Fig. 7). Our results supported the feasibility of FRTACs on depleting protein target in vivo, which is consistent with our in vitro results. The elevation of CD8+ T cell infiltration in tumor tissues from mice treated with the degrader than those with antibody suggested that the FRTAC could promote the penetration of more cytotoxic T lymphocytes into the tumor to induce the tumor killing effect, therefore suppressing tumor growth more significantly than the antibody. Overall, our results indicate that the FRTAC targeting PD-L1 is more effective immunotherapy than the PD-L1 blocker in all three models.

### Cancer selectivity of FRTACs targeting EGFR and PD-L1

Lastly, to demonstrate that FRTAC can selectively degrade the protein targets in cancer cells, thereby reducing the on-target/off-tumor effect in normal cells, we compared the degradation efficiency of two FRTACs, Ctx-FA and Atz-FA, in a non-cancerous cell (human keratinocyte line HACAT) with selected cancer cells expressing similar basal EGFR or PD-L1 expression level as the HACAT cells, respectively (Supplementary Fig. 8a). Our results showed that no significant degradation of EGFR was observed in HACAT cells, while the level of EGFR was dramatically reduced in Huh7 cells. Similarly, Atz-FA induced significant degradation of PD-L1 in Huh7 and TU138 cells, whereas minimal degradation of PD-L1 was detected in HACAT cells. (Fig. 6a,b). Western blot showed significantly lower expression level of FR in HACAT cells compared to the cancer cell lines, which may contribute to the much lower degradation efficiency of both FRTACs on normal cell lines (Fig. 6c). Next, we assessed the cancer selectivity in vivo by characterizing the morphology and detecting PD-L1 level on normal tissues in the CT26 syngeneic mouse model under the conditions indicated in Fig. 6d. H&E staining showed that no obvious morphological changes in spleen and lung were observed in the degrader-treated mice. Moreover, the treatment of Ab3-FA resulted in no PD-L1 degradation in spleen and lung, as evidenced by both western blot and IHC staining results, which contrasted with the significant reduction of PD-L1 detected in tumor tissues (Fig. 6e and Supplementary Fig. 8b, c). Overall, these data indicate the potential of FRTACs for selective degradation of the protein targets in cancer cell lines.

### Discussion

In summary, we demonstrated that FR could serve as a LTR for the efficient degradation of both soluble and membrane protein targets by chimeric FRTACs. FRTACs were shown to leverage both isoforms of FR to transport the protein targets into the lysosome for depletion without impeding the recycling of the receptors. In addition to factors such as the degree of FA labeling and linker length, we found that the degradation efficiency also highly depends on the cellular FR expression level for extracellular soluble proteins. Other parameters, such as

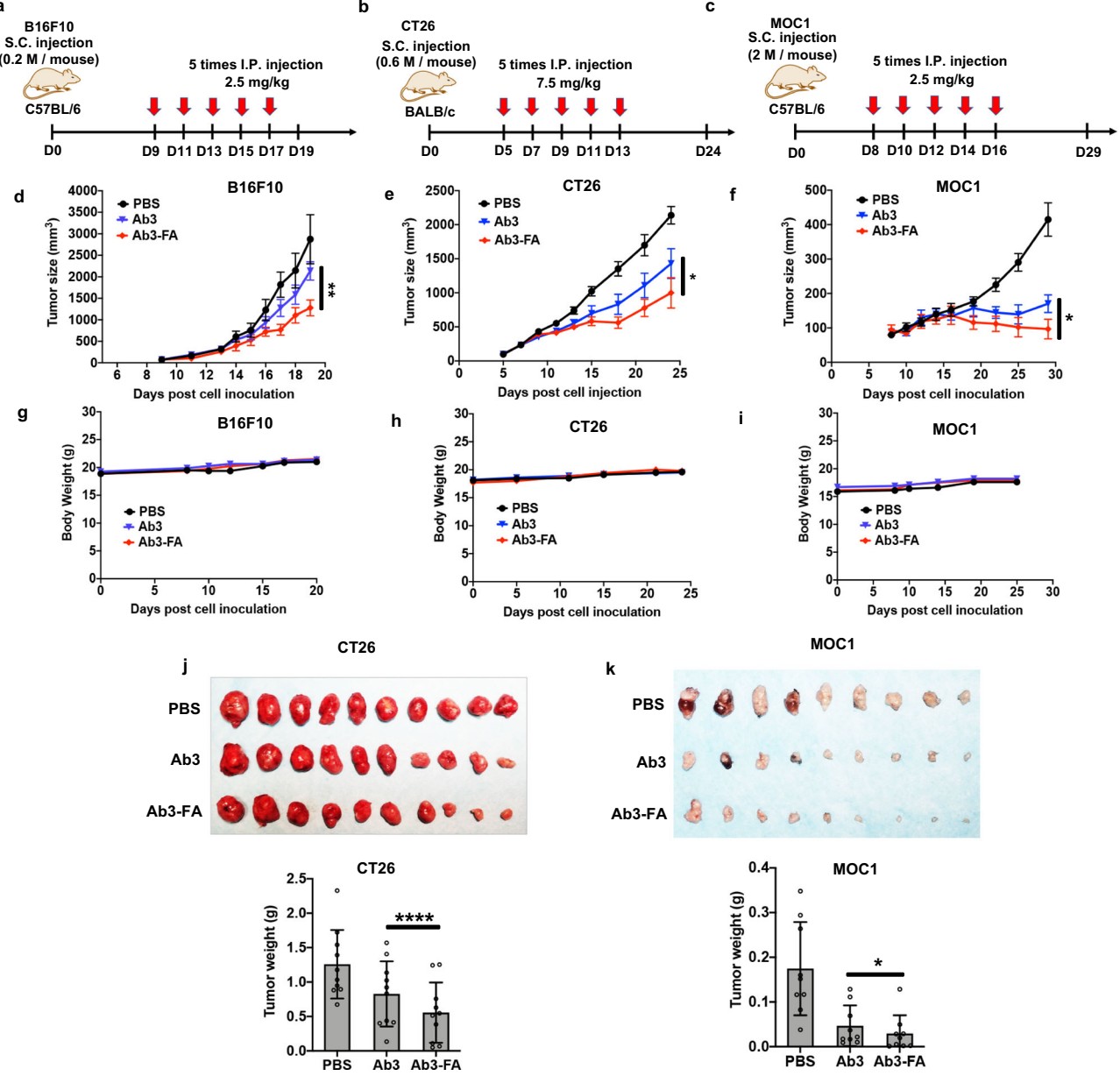

**Fig. 5 | FRTAC targeting PD-L1 inhibits tumor growth in B16F10, CT26 and MOC1 syngeneic mouse models. a** Schematic illustration of FRTAC treatment in B16F10 mouse model. **b** Schematic illustration of FRTAC treatment in CT26 mouse model. **c** Schematic illustration of FRTAC treatment in MOC1 mouse model. **d** Tumor growth curves after different treatments as indicated by (**a**) (n = 8). **e** Tumor growth curves after different treatments as indicated by (**b**) (n = 10). **f** Tumor growth curves after different treatments as indicated by (**c**) (n = 9). **g.** Body weight curves after different treatments as indicated by (**a**) (n = 8). **h** Body weight curves after different treatments as indicated by (**b**) (n = 10). **i** Body weight curves after different treatments as indicated by (**c**) (n = 9). **j** Images and weight of excised tumors on day 24 after different treatments as indicated by (**b**) (n = 10). Data are presented as mean ± SD, n = 10. **k** Images and weight of excised tumors on day 29 after different treatments as indicated by (**c**) (n = 9). N indicates mice. Data are presented as mean ± SD. The statistical significance was assessed using a paired one-tailed t test for (**d–f**) and a paired two-tailed t test for (**j, k**), *P < 0.05, **P < 0.01, ****P < 0.0001. Source data are provided as a Source Data file.

the expression of the endogenous protein target and potential interactions between the FR and protein target induced by FRTACs, may also play roles in the degradation efficiency. Degraders exhibit lower degradation efficiency for membrane proteins in cells expressing abundant amounts of target protein but low levels of FR. This is likely due to insufficient FR-mediated targeting, leading to suboptimal engagement of the degraders with the target protein. The ratio between FR and endogenous target expression levels is a better determinant of membrane protein degradation compared to FR expression alone. A series of FRTACs was generated and validated for their capability of depleting corresponding targets with different structures and functions, including mouse IgG, EGFR, PD-L1 and CD47,

in a variety of human and mouse cancer cell lines. We demonstrated that FRTAC targeting PD-L1 exhibited similar clearance rate but superior in vivo anti-tumor effects than the therapeutic antibody in B16F10, CT26, and MOC1 tumor models. This tumor killing effect of FRTAC could be attributed to the reduction of tumorous PD-L1 and the increased cytotoxic T cells in the tumor. Finally, a much more significant degradation of EGFR and PD-L1 observed in cancer cells over normal cells, as well as no degradation of PD-L1 in normal tissues after FRTAC treatment in mice indicate the potential for lower toxicity and high cancer selectivity of FRTACs.

To date, only a limited number of LTRs have been explored for the development of degraders of extracellular proteins. Most of these

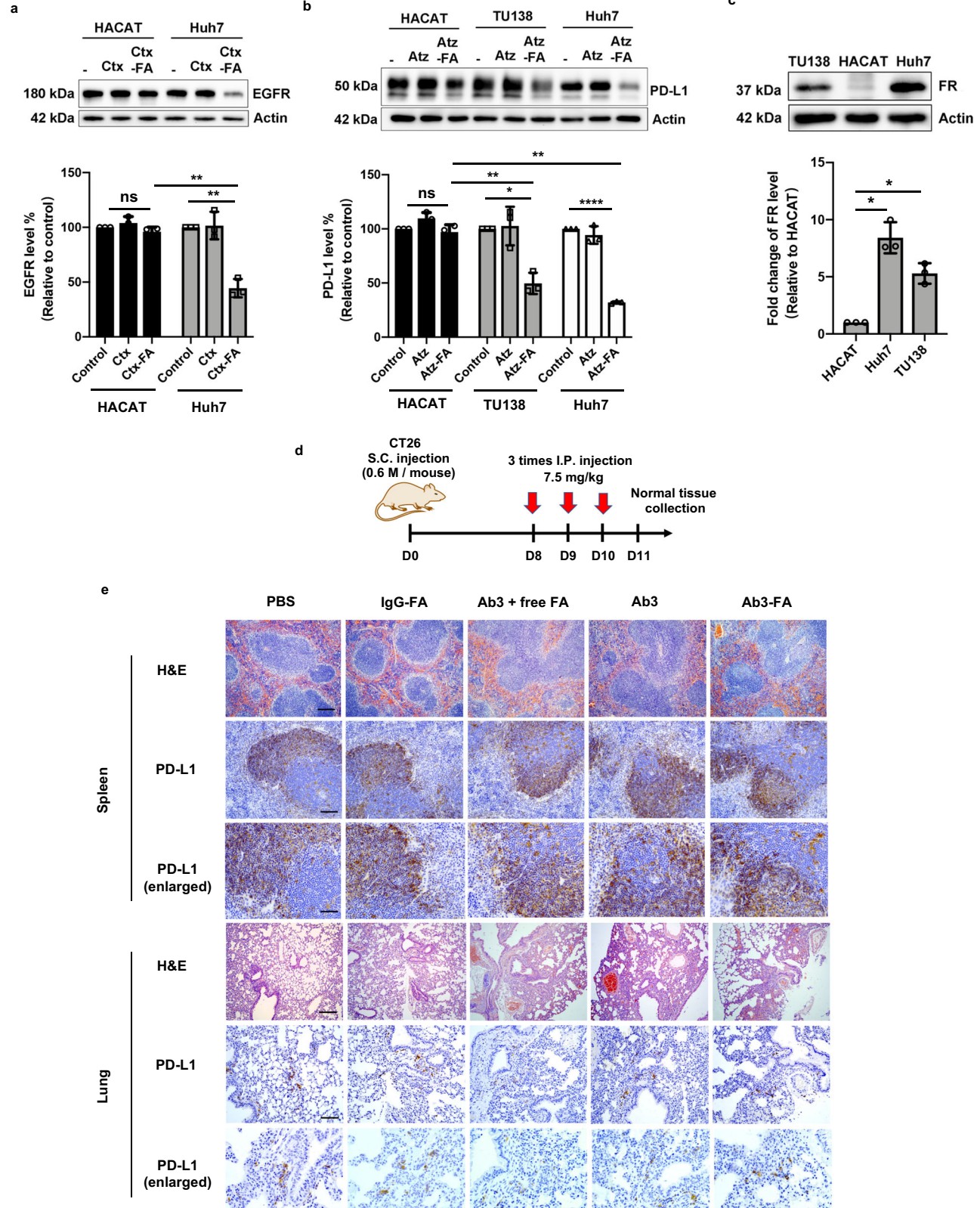

**Fig. 6 | FRTACs show cancer selectivity for the degradation of EGFR and PD-L1.**
**a** Comparison of EGFR degradation efficiency mediated by Ctx-FA (10 nM) in normal cell line HACAT, and cancer cell line Huh7 (n = 3). **b** Comparison of PD-L1 degradation efficiency of Atz-FA in normal cell line, HACAT, and cancer cell line Huh7 and TU138 (n = 3). **c** Quantification of FR expression levels on HACAT, Huh7, and TU138 by western blot (n = 3). **d** Schematic illustration of FRTAC treatment in CT 26 mouse model. **e** Characterization of spleen and lung morphology by H&E

staining and detection of PD-L1 level in spleen and lung after FRTAC treatment by IHC staining. Scale bar for H&E staining and PD-L1: 500 μm, Scale bar for PD-L1 enlarged: 100 μm. Scale bar for PD-L1 enlarged: 50 μm. Representative figures from three mice. N indicates biologically independent experiments. Data are presented as mean ± SD. The statistical significance was assessed using an unpaired two-tailed $t$ test, *P < 0.05, **P < 0.01, ****P < 0.0001, ns: not significant. Source data are provided as a Source Data file.

LTRs are ubiquitously expressed in many types of cells and tissues with the exception of ASGPR, which is primarily expressed in the liver. Degraders that can recruit LTRs primarily expressed in relevant pathogenic tissues would significantly reduce potential toxicity issues caused by on-target/off-tissue effect. Although integrin and transferrin receptor have been employed as LTRs[25,26], no cancer selectivity was demonstrated. Other recycling receptors with selective expression pattern may contribute to the tissue specific degraders, such as follicle-stimulating hormone receptor and prostate-specific membrane antigen[55,56].

The development of FRTAC expands the landscape of LTR that can be used for degraders of extracellular secreted and membrane proteins. The readily available FR ligand represents a significant advantage over existing lectin type of LTRs that require complex polymeric or oligomeric carbohydrate ligands. More importantly, in addition to the overexpression of FR in various cancers, the inaccessibility of FR in normal tissues to the circulating FA conjugates, a characteristic not shared by other cancer-overexpressing receptors, enables selective degradation of cancer-relevant protein targets in cancer cells. Indeed, higher degradation efficiency for targets in malignant cells/tissues than normal cells/tissues suggests that on-target/off-tumor effect can be minimized using FRTACs. Furthermore, the successful degradation of multiple therapeutically relevant protein targets in a number of different cancer cell lines indicates that the scope of FRTAC on depleting various cancer-associated protein targets is broad. Beyond the targets examined in this paper, FRTAC can also be applied to various other cancer-relevant targets to reduce the on-target/off-cancer toxicity associated with current therapeutic inhibitors[57-62]. These include matrix metalloproteinases, heparanase and pH regulator carbonic anhydrase IX[63-65]. In addition to the antibody-based FRTACs, the possibility of transporting the soluble protein targets into the cells by small molecule FRTAC has also been demonstrated. Higher uptake efficiency of antibody-based FRTAC over small molecule FRTAC was observed, which is likely due to the multivalency of the antibody-based degrader.

Immune checkpoint blockade (ICB) therapy against PD-L1 is hindered by the low patient response rate and toxicity issues due to non-selective recognition of PD-L1 in normal tissues. As indicated in three mouse models with distinct response efficacy to PD-L1 inhibition and degradation, our FRTAC led to more prominent delayed tumor progression and elicited a stronger immune response than the therapeutic antibody. The modification of FA on the antibody only slightly altered the circulating time in vivo. Taken together, FRTACs have the potential to become a more effective cancer immunotherapy than the blocking antibody by increasing the response rate and reduce the on-target/off-tumor side effect.

Collectively, our study demonstrated the feasibility of a FR-mediated cancer-selective TPD platform and uncovered a potential therapeutic alternative for ICB therapy with improved precision and efficiency. This work will add an additional dimension to TPD with cancer cell selectivity and FRTAC could be a general, effective, and easily accessible technology to degrade cancer-associated protein targets.

## Methods
### Cell culture
HepG2 and Huh7 cells were cultured in low-glucose DMEM supplemented with 10% fetal bovine serum, 1% non-essential amino acids, 1% sodium pyruvate, 1% L-glutamine and 1% penicillin/streptomycin under 5 % $CO_2$ at 37 °C. Hela, Fadu, SW480, Cal27, TU138, HACAT, and MDA-MB-231 cells were maintained in high-glucose DMEM supplemented with 10% fetal bovine serum, and 1% penicillin/streptomycin under 5 % $CO_2$ at 37 °C. CT26, B16F10, MOC1 and A549 cells were cultured in RPMI supplemented with 10% fetal bovine serum, 1% sodium pyruvate,

1% HEPES and 1% penicillin/streptomycin under 5 % $CO_2$ at 37 °C. B16F10, HepG2, Huh7, Hela, MDA-MB-231 and A549 were obtained from the American Tissue Culture Collection (ATCC). Fadu, SW480, Cal27, HACAT, CT26, MOC1, TU138 were obtained from UW-Madison Head Neck Cancer SPORE Pathology Core.

### Cellular uptake of anti-FITC and anti-biotin
Cells were seeded at 70% confluence and maintained in 200 μL complete culture media in a 48-well plate. The next day, cells were treated sequentially with 25 μL medium containing anti-FITC or anti-biotin and 25 μL medium containing PBS, antibody, or degrader at different concentrations. The cells were incubated at 37 °C for different time periods and then washed twice with PBS before being harvested for in-gel fluorescence analysis.

### Anti-biotin degradation analysis
Cells were seeded at 70% confluence and maintained in 200 μL complete culture media in a 48-well plate. The next day, cells were incubated with 25 nM of Ab-FA and 50 nM of anti-biotin-647 for 3 h, followed by three washes with PBS. Cells were maintained subsequently in fresh media with or without 20 μM chloroquine (CQ), 200 nM Bafilomycin A1 (BAF), and 1 μM MG132 for another 24 h before being harvested for in-gel fluorescence analysis.

### FR1 knockdown
Hela cells were seeded at 200,000 cells per well in a 6-well plate one day before transfection. Cells were transfected with 60 pmol of scramble or FR1 siRNA and 7.5 μL RNAiMAX for 24 h and then re-seeded in a 48-well plate at a density of 70,000 cells per well. Cells were then incubated with 50 nM Ab-FA and 50 nM anti-Rabbit-647 for another 6 h before harvested to analyze FR1 levels by western blot and the uptake of anti-rabbit-647 by in-gel fluorescence.

### FR1 and FR2 overexpression
Hela cells were seeded at 70,000 cells per well in a 24-well plate one day before transfection. Cells were transfected with 140 ng plasmid expressing FLAG-FR1 or FLAG-FR2 and Fugene transfection reagent at 1:3.5 w/v ratio for 48 h. Cells transfected with the same amount of empty vectors were used as a negative control. After replacing the media containing the transfection reagent and plasmid with fresh media, cells were incubated with 50 nM Ab-FA and 50 nM anti-Rabbit-647 for another 6 h. Then cells were harvested to analyze FLAG-FR1 and FLAG-FR2 expression levels by western blot and the uptake of anti-rabbit-647 by in-gel fluorescence.

### Quantification of cell surface FR level
The expression of cell surface FR was analyzed by flow cytometry. Briefly, flow buffer was prepared as 2% FBS in PBS. $4 \times 10^5$ cells were harvested, washed with PBS, and then re-suspended in 400 μL ice-cold flow buffer containing 10 μM folate-FITC to incubate for 1 h at 37 °C. Cells were then washed twice with 1 mL ice-cold flow buffer and resuspended in 500 μL flow buffer with 1 μg/mL DAPI before flow analysis. Flow cytometry data were acquired using ThermoFisher Attune NxT cytometric software (v 6). FlowJo (v 10.8.1) was used for flow cytometry data analysis.

### Antibody labeling with FA-PEGn-NHS (1k- or 2k-PEG-linker)
To label the antibody with folate, 100 μL of the antibody (concentration at 1.8 mg/mL) in PBS was mixed with folate-PEGn-NHS ester at 1:3, 1:12, or 1:25 molar ratio. The reaction was incubated overnight at room temperature on a rotator, followed by filtration with 500 μL of PBS 5 times using a 10 kDa Amicon Centrifugal Filter. The mass of unmodified antibody and FA-labeled antibody was analyzed by MALDI-MS, and the average number of FA per antibody was calculated as below

(Supplementary Table 3):

$$Ave.NO\ of\ FA\ per\ Ab = \frac{Mass\ of\ (FA-antibody) - Mass\ of\ antibody}{Molecular\ weight\ of\ (FA-PEGn-NHS)}$$

### Antibody labeling with DBCO-PEG3-NHS and Folic acid-PEGn-N3 (1k- or 2k-PEG-linker)

To label the antibody with DBCO, 200 µL of the antibody (concentration at 1.8 mg/mL) in PBS was mixed with DBCO-PEG3-NHS ester at a 1:25 molar ratio. The reaction was incubated overnight at room temperature on a rotator, followed by filtration with 500 µL of PBS 5 times using a 10 kDa Amicon Centrifugal Filter. Then, the concentration of DBCO-labeled antibody was determined by BCA assay and mixed with Folic acid-PEG1k-N3 or Folic acid-PEG2K-N3 at a 1:25 molar ratio. The reaction was incubated overnight at room temperature on a rotator, followed by filtration with 500 µL of PBS 5 times using a 10 kDa Amicon Centrifugal Filter. The mass of unmodified antibody, antibody-DBCO, and FA-labeled antibody was analyzed by MALDI-MS, and the average number of FA per antibody was calculated as below (Supplementary Table 3):

$$Ave.NO\ of\ FA\ per\ Ab = \frac{Mass\ of\ (FA-antibody) - Mass\ of\ (DBCO-antibody)}{Molecular\ weight\ of\ (FA-PEGn-N_3)}$$

### MALDI-MS

Part of the samples were characterized by the following method: Matrix solution was made by dissolving α-Cyano-4-hydroxycinnamic acid (CHCA) in 50% Acetonitrile/H$_2$O at a final concentration of 10 mg/mL. The sample was absorbed on Omix C4 pipette tips, followed by washing with 0.1% TFA three times and then eluted with 20 µL 75% Acetonitrile/H$_2$O. 1 µL sample solution and 1 µL CHCA solution were spotted on the MALDI target plate and mixed thoroughly before the spot was allowed to dry at room temperature. MALDI-MS spectra were acquired on the Bruker UltraFlex MALDI-TOF/TOF mass spectrometer operated in linear positive ion mode and plots were generated by Bruker flexAnalysis 4.2. The rest of the samples were characterized by the following method: Matrix solution was made by dissolving sinapic acid (SA) in 70% Acetonitrile/H$_2$O with 0.1% TFA at a final concentration of 20 mg/mL. The sample was absorbed on Omix C4 pipette tips, followed by washing with 0.1% TFA five times and then eluted with 15 µL 75% Acetonitrile/H$_2$O. After desalting the sample with Omix C4 pipette tips, 1 µL sample solution and 1 µL SA solution were spotted on the MALDI target plate and mixed thoroughly before the spot was allowed to dry under room temperature. MALDI-MS spectra were acquired on Bruker RapifleX MALDI TOF mass spectrometer (Bruker Scientific, LLC, Bremen, Germany) operated in linear positive ion mode. The smartbeam laser was set to 100% with few thousand shots (labeled on each of the spectrum) per spot at a repetitive rate of 1000 Hz, and detector gain was set at 600 V for the experiments after method optimization. Spectra were processed in Bruker flexAnalysis 4.2 (Supplementary Fig. 10).

### Targeted protein degradation

Cells were seeded at 70% confluence in a 24-well plate one day before treatment. Then, cells were treated with FRTACs targeting EGFR, PD-L1, or CD47 at various concentrations for indicated time periods before collection for western blot analysis. For mouse PD-L1, B16F10, CT26 and MOC1 cells were pre-incubated with 100 ng/mL mouse IFNγ for 24 h to induce PD-L1 expression before degrader treatment. For human PD-L1, Huh7, HACAT, and TU138 cells were pre-incubated with 100 ng/mL human IFNγ for 16 h to induce PD-L1 expression before degrader treatment.

### Rab7 knockdown

Hela cells were seeded at 200,000 cells per well in a 6-well plate one day before transfection. Cells were transfected with 100 pmol of scramble or Rab7 siRNA and 7.5 µL RNAiMAX for 48 h and then re-seeded in a 48-well plate at a density of 70,000 cells per well. To test the uptake of soluble protein, cells were then incubated with 50 nM Ab-FA and 50 nM anti-Rabbit-647 for another 3 h before being harvested to analyze Rab7 levels by western blot and the uptake of anti-rabbit-647 by in-gel fluorescence. To test EGFR degradation, cells were treated with 10 nM Ctx-FA for 6 h before being harvested for western blot analysis.

### Western blotting and in-gel fluorescence analysis

Cells were lysed in 1x RIPA lysis buffer (25 mM Tris, pH 7–8, 150 mM NaCl, 0.1% (w/v) sodium dodecyl sulfate (SDS), 0.5% sodium deoxycholate, 1% (v/v) Triton X-100, protease inhibitor cocktail (Roche, one tablet per 10 mL)) on ice for 10 min, followed by centrifugation at $16,000 \times g$ at 4 °C for 15 min. The supernatant was collected and adjusted to the equal amounts after determining the protein concentration using the BCA assay. Lysates were then mixed with the 4x Laemmli Loading Dye and heated at 99 °C for 5 min before being loaded onto 7.5 or 12% SDS−polyacrylamide gel electrophoresis. For western blotting, the gel was transferred to a PVDF membrane, blocked in 5% (w/v) nonfat milk in the TBST washing buffer (137 mM NaCl, 20 mM Tris, 0.1% (v/v) Tween) and then incubated with primary antibodies at 4 °C overnight. After 3 washes with TBST, the membrane was incubated with secondary HRP-linked antibodies for 1 h, and then washed 3 times with TBST. Then the membrane was incubated in the Clarity ECL substrate for 3–5 min before acquiring the immunoblot by ChemiDoc MP Imaging Systems. For in-gel fluorescence analysis, the fluorescence images of the gel were directly acquired by ChemiDoc MP Imaging Systems and the total protein was detected using Coomassie blue staining. Western blot and in-gel fluorescence images were acquired by Image Lab Touch Software (v 2.0.1.35). Western blot bands intensity was analyzed using ImageJ (v 1.53a).

### Confocal microscopy

Hela cells were seeded onto 8-well chamber slides at density of 30,000 cells per well in 200 µL of complete culture medium. After adhesion, cells were treated with 50 nM Ab-FA and 50 nM anti-Rabbit-647 for 24 h at 37 °C, followed by the 30-min incubation with Lyso-Tracker Green DND26 (100 nM) at 37 °C. Hoechst 33342 (5 µg/ml) was added 10 min before the end of incubation. After three washes with PBS, the live cells were imaged using Leica SP8 3X STED super-resolution microscope at 20x magnification with a 10x eyepiece and analyzed by ImageJ. The Pearson's correlation coefficients and mean fluorescence intensity (MFI) were analyzed by Leica LAS-X software (v 2.6).

Fadu cells were seeded onto 8-well chamber slides at a density of 20,000 cells per well in 200 µL of complete culture medium. After adhesion, cells were treated with 10 nM of Ctx-FA for 24 h at 37 °C. Cells were then washed with PBS for 3 times and fixed with 4% paraformaldehyde for 15 min followed by permeabilization with 0.5% Triton X-100 for 5 min. After blocking with 5% BSA for 1 h at room temperature, the cells were co-incubated with anti-LAMP1 antibody in 1% BSA overnight at 4 °C. The next day, cells were washed with PBS and then incubated with anti-mouse-488 and anti-rabbit-594 secondary antibodies for 1 h at room temperature. Then the cells were mounted with slowfade-antifade mounting medium containing DAPI after three washes. Images were acquired by Leica SP8 3X STED super-resolution microscope at 60x magnification with a 10x eyepiece and analyzed by ImageJ. The Pearson's correlation coefficients and mean fluorescence intensity (MFI) were analyzed by Leica LAS-X software (v 2.6).

## Animal studies

All animal work were approved by the University of Wisconsin-Madison Institutional Animal Care and Use Committee (IACUC) and conducted in accordance with the NIH Guide for the care and use of laboratory animals (animal protocol number M006396). Mice that were 6–8 weeks of age were purchased from the Jackson Laboratory. Mice were group-housed, with no more than five mice per cage. Environmental conditions were maintained at a temperature of 22–25 °C and a humidity level of 40–60%. The mice were kept on a 12-h light/dark cycle with ad libitum access to food and water.

## Pharmacokinetics (PK) analysis

The pharmacokinetics of antibody and degrader were determined by intraperitoneally injecting 2.5 mg/kg of antibody and FRTAC into C57BL/6 mice or C57BL/6 mice bearing B16F10 tumor. A small amount of blood was collected from mouse tail vein at different time points and the corresponding levels of antibody and FRTAC were measured by blotting the Rat IgG level in the plasma. Ab3-FA-25x at different concentrations were used as standards.

## Tumor growth experiments

B16F10 ($2 \times 10^5$ cells/mouse to C57BL/6), CT26 cells ($6.0 \times 10^5$ cells/mouse to BALB/c) and MOC1 cells ($2 \times 10^6$ cells/mouse to C57BL/6) were inoculated in mice subcutaneously. When tumor volumes reached 50–150 mm$^3$, C57BL/6 mice (n = 8/cohort) bearing B16F10 tumor were treated with PBS, Ab3 or Ab3-FA at a dose of 2.5 mg/kg via IP injection, and continuously treated every other day for a total of 5 doses. When tumor volumes reached 50–150 mm$^3$, BALB/C mice (n = 10/cohort) bearing CT26 tumor were treated with PBS, Ab3 or Ab3-FA at a dose of 7.5 mg/kg via IP injection, and continuously treated every other day for a total of 5 doses. For control groups, BALB/C mice (n = 6/cohort) bearing CT26 tumor were treated with PBS, IgG-FA, Ab3 or Ab3 + free FA at a dose of 7.5 mg/kg via IP injection, and continuously treated every other day for a total of 5 doses. When tumor volumes reached 20–100 mm$^3$, C57BL/6 mice (n = 9/cohort) bearing MOC1 tumor were treated with PBS, Ab3 or Ab3-FA at a dose of 2.5 mg/kg via IP injection, and continuously treated every other day for a total of 5 doses. The mouse body weight was monitored and tumor volume size was measured every 2 days with a caliper and calculated with the formula 0.5 x length × (width)$^2$.

## Evaluation of PD-L1 and CD8a level in vivo

CT26 cells ($6.0 \times 10^5$ cells/mouse to BALB/C) were inoculated in mice subcutaneously. When the tumor volume reached to 100–200 mm$^3$, the mice were treated with PBS, IgG-FA, the combination of Ab3 plus free FA (0.11 mg/kg), Ab3, or Ab3-FA (7.5 mg/kg daily for 3 days, 3 mice/cohort). Half of tumors, spleen, and lung from CT26 syngeneic mouse model were frozen and lysed in RIPA buffer for western blot analysis. The remaining tumors, spleen, and lung were fixed in formalin, and embedded in paraffin. Paraffin-embedded tumor and normal tissue specimens were cut into 5-μm sections for immunohistochemistry (IHC) staining.

## Immunohistochemistry (IHC) staining

In brief, tumor sections were deparaffinized and rehydrated, followed by antigen retrieval in citrate buffer (10 mmol/L, pH 6.0) for 15 min at 98 °C. After quenching with 3% hydrogen peroxide and blocking with 2.5% goat serum, slides were incubated at 4 °C overnight with primary antibodies against PD-L1 or CD8 followed by a 30-min incubation with HRP-conjugated secondary antibody. Slides were then visualized using the Dakocytomation Liquid DAB+Substrate Chromogen System. After counterstaining with hematoxylin, the tumor sections were dehydrated and mounted with coverslips.

## Reporting summary

Further information on research design is available in the Nature Portfolio Reporting Summary linked to this article.

## Data availability

All data supporting the findings of this study are available within this manuscript and its Supplementary Information files. Specific data $P$ values are included within the Source Data file. Source data are provided with this paper.

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

## Acknowledgements

W.T. thanks financial support for this research from the University of Wisconsin-Madison Office of the Vice Chancellor for Research and Graduate Education with funding from the Wisconsin Alumni Research Foundation through a UW2020 award and NIH R01 CA284689. This study made use of the Medicinal Chemistry Center and flow cytometry facility, supported by the University of Wisconsin Carbone Cancer Center Support Grant NIH P30 CA014520. L.L. acknowledges funding support of NIH shared instrument grants (NIH-NCRR S10RR029531, S10OD028473 and S10OD025084), NIH grants R01 DK071801, R01 AG052324, and R01AG078794, a Vilas Distinguished Achievement Professorship, and Charles Melbourne Johnson Professorship with funding provided by the Wisconsin Alumni Research Foundation and University of Wisconsin-Madison School of Pharmacy. We would also like to thank the University of Wisconsin Optical Imaging Core for the use of the Leica SP8 3x STED super-resolution microscope. Lastly, we would like to thank staff in Research Resource Center for breeding the mice for the animal studies.

## Author contributions

Y. Zhou and W.T. conceived the project. Y. Zhou performed most in vitro biological experiments and interpreted the data. C.L. performed most in vivo mouse experiments and interpreted the data. X.C., Y. Zhao, Y. L., and N.N. assisted with biological and animal experiments. P.H., W.W., and L.L. conducted and provided advice on the MALDI-MS analysis. Y. Zhou and W.T. wrote the manuscript with input from all authors.

## Competing interests

A patent (WO2023183381A3, Inventors: Weiping Tang, Yaxian Zhou, Chunrong Li, and Hao Wu) related to this manuscript has been filed by Wisconsin Alumni Research Foundation, which acts as the technology transfer entity for the University of Wisconsin-Madison. The patent has been published. The use of folate as the ligand of lysosome targeting receptor to develop FRTACs for degrading disease-associated proteins is covered in the patent application. W.T. is the co-founder and shareholder of Chimergen Therapeutics Inc. The remaining authors declare no competing interests.
