## [Peer Review File · Nature Communications]

Reviewers' Comments:

Reviewer #1:

Remarks to the Author:

Targeted protein degradation (TPD) technology is an emerging concept in recent years. Since the core of the technology is to use the cell's own degradation mechanism to remove target proteins, in principle, there is almost no target limitation for TPD. However, TPD technology relies on the developer's rational use of the cell's own degradation pathways. Proteolysis Targeting Chimera (PROTAC) has been developed based on the E3 ubiquitin ligase-dominated degradation pathway. Lysosome Targeting Chimeras (LYTACs) have also been developed based on the cell's own lysosomal degradation pathway. In this manuscript, the authors creatively propose the use of folate receptor (FR)-mediated endocytosis for the degradation of cancer-associated extracellular and membrane proteins. Specifically, FR, a major biomarker for a variety of tumors, mediates the efficient endocytosis of folate and folate conjugates by cells. Taking advantage of this property, the authors used click chemistry to sequentially couple folate to antibodies to different proteins and characterized the degradation of FRTAC at extracellular soluble and endogenous membrane proteins. Based on this, the authors used PL-L1 as a target to reduce the protein amount of PD-L1 in mice by FR as a way to achieve anti-tumor effects. Thus, the use of this technology greatly expands the means by which TDP technology can degrade extracellular and membrane proteins, and is of great value in assisting in the treatment of tumors.

The novel TPD system developed by the authors using the new technology pathway is certainly exciting. However several points should still be made to further improve this work.

Key points:

- 1, Regarding the cellular imaging present in the manuscript (Fig.2d; Fig.2e), the authors should consider quantifying the amount of EGFR remaining as well as the co-localization of EGFR with lysosomes.
2. Regarding the multiple experiments in the manuscript (Fig.2d-h; Fig.4J-k; Fig.5f), it is necessary for the authors to include FA controls in the experimental design to ensure that the results are not due to the presence of FA. Alternatively, the target Ctx could be replaced with a protein with no target sequence and coupled with FA as a control. This will further make the experimental conclusion more credible.
- 3, Regarding Fig.3F, the authors suggest that the addition of BAF can effectively inhibit Ctx-FA-mediated EGFR degradation. However, it seems that the inhibition effect of BAF

is significantly less effective compared to (Fig.S4I; Fig.4j) or the FA over-addition in Fig.3F. The authors need to further discuss the presentation of this result with a revised description of the conclusions. Are there other downstream pathways that dominate FR-mediated degradation.

4, Regarding the presentation of the results of Fig.4J-K, the authors performed mouse tumor formation experiments with B16F10, CT26, MOC1, but only the results of CT26 and MOC1 were presented. the presentation of the results of B16F10 is necessary. Also as mentioned in #2, the experiment needs to add further reasonable controls.

5. Regarding the demonstration of the tumor inhibition effect of Ab3-FA in Fig.4, it seems that the tumor inhibition effect of Ab3-FA is not greatly enhanced compared to the Ab3 use group. Is it possible to add more positive control and negative reference to this experiment to better evaluate the effect of FRTACs.

6, The authors state in Fig.S5a-b that the effect of Ab3-FA was a decrease in PD-L1 expression as well as an increase in the infiltration of CD8a+ cells in the tissue. Whether the use of antibodies to CD8a in this use could antagonize the tumor inhibitory effect of Ab3-FA. Also compared to the results of Fig.4J-K, the rise in the number of CD8a+ cells is significant, how to understand the results of this experiment that the tumor inhibitory effect is not strong.

7、 Besides PD-L1, what are the other applicable targets for FRTACs? The authors have a powerful system for degradation of extracellular and membrane proteins, can they find more suitable targets for this technology to achieve better inhibition of tumor progression?

8. The authors have developed a set of protein degradation tools based on endocytosis of cell surface proteins. In addition to the excellent endocytosis receptor FR of tumor cells, the authors should discuss whether other endocytosis receptor pathways can be used to expand the use of this technology.

Reviewer #2:

Remarks to the Author:

N.B.: Word document with same information (but better formatting) is attached.

NCOMMS-23-61763 Review “Development of Folate Receptor Targeting Chimeras (FRTACs) for Cancer Selective Degradation of Extracellular Proteins”

The authors of this manuscript report on the development of Folate Receptor Targeting Chimeras (FRTACs) which are a new modality for extracellular targeted protein degradation (ETPD) which is an up-and-coming area of research. The novelty of the work lies in the new degradation-inducing ligand (folic acid, FA) and the concept of FRTACs being selective for cancer. While there is another recent example of a cancer-selective ETPD moiety (which targets integrins, Zheng et al., JACS, 2022) that work did not examine the selectivity of their reagent for cancer vs healthy tissues. Also important to note is that FA has been used as a cancer-targeting agent for PROTACs (referenced in the manuscript) in a strategy that relied on FA-mediated internalization, but this application is certainly different enough to merit novelty.

The work reported in this manuscript is of high quality and rigor. Appropriate controls are included and FRTACs are examined from multiple different angles. The results are promising, with a PD-L1-FRTAC showing increased in vivo efficacy compared to PD-L1 blockade by the parent antibody at the same concentration. PD-L1-FRTAC also showed lower degradation of PD-L1 in healthy tissue than in cancer cells, suggesting it is indeed selective, which could very well translate to higher safety – although this wasn't conclusively demonstrated here. I think the data showing that the ratio of target:lysosomal targeting receptor (LTR) is a key determinant of ETPD efficacy is a key finding as well, which could bear more highlighting in the manuscript. Perhaps this is generalizable to other ETPD systems and would be useful to bear in mind for others working in the field. The manuscript is well-written and easy to follow in general, though it would benefit from some additional proofreading. I have provided line-by-line points below, but these are relatively minor issues.

I do, however, have some issues with the MS data. Most of the MS spectra are up to standard, but in some of them the signal-to-noise ratio is so low, I do not think meaningful conclusions can be drawn.

These are: *S7a/Ab-FA*, *S7b/NHS_12x (PEG2k)*, *S7d/Atz-FA*, S7e/Ab2-FA, S7f/Ab3 (a native antibody should look better!), *S7f/Ab3-FA-12x*, *S7f/Ab3-25x*. I have starred the worst offenders, and these need to be repeated. I think repeating the two unstarred ones would also improve the manuscript, but these are just about acceptable as they are.

Your MALDI data in your 2021 ACS Central Science paper are a good example of good MALDI spectra as well as those from this manuscript I have not highlighted. I do understand that it is hard to get clear MS data for heterogeneous NHS-ester modified conjugates, but if you are unable to get meaningful data this way, a different form of analysis is required. Some form of HPLC perhaps? SEC or HIC? These would give less precise data, but at least you could get better signal-to-noise and have a better chance of actually interpreting it.

In summary, I would recommend accepting this manuscript provided the above MALDI data are improved, or additional analysis is used to characterize the highlighted conjugates in a satisfactory manner, and if my below comments are addressed.

Sincerely,

Peter Szijj

Specific line-by-line comments:

Line 11 – target through the cell’s own disposal systems

Line 15 – which would

Line 20 – Look into formatting of in vivo and in vitro

Line 29 – PROteolysis TArgeting Chimeras (PROTACs) were developed

Line 30 - Over a dozen of PROTACs that

Line 32 - In addition, about a dozen of other

Line 33 - demonstrated in their utility

Some language editing would be needed (will not highlight from now on). N.B.: The rest of the manuscript has a lower density of errors, but some additional proofreading would be beneficial.

Line 35 – I think you should talk about the FR-targeted PROTACs a little, as that is relevant and, in some ways, similar to this work (but certainly different enough!)

Line 52 – Integrins are overexpressed on various cancers, thus the integrin-based LYTACs in your Ref 27 (Zheng et al. 2022) do represent selective degradation in a disease setting. I think this is a key reference that you should dwell on in more detail. If your work has advantages over this, either by virtue of folate-targeting or through experimental design, you can detail that here. I would highlight that your work actually examined the selectivity of the ETPD strategy for cancer vs. healthy tissue, which the integrin-paper did not.

Line 145 – I would show the data from Fig S1D in Fig 1 as well. They complement each other well and it is good to have a side-by-side comparison

Line 168 – Any comment on why the 2-step protocol worked better? I cannot think of any obvious explanation if the same degree of labelling is achieved in both cases

Line 172 – Any comment on why 2k-PEG was better than 1k-PEG? Longer length allowing dual engagement of both receptors better?

Line 750 – Cell type (Fadu, right?) not noted in Fig caption for B-G

Line 206 – It’s an interesting concept that the ratio of EGFR:FR is what really matters, and it is great that you have established this. You could explain this in more detail in the text, as I think this is a key point for ETPD in general!

Line 246 – I would not use the term “significantly” without backing it up statistically. I would also argue that a 3-fold decrease is notable, and could be attributable to the random nature of lysine-conjugation (i.e., lysines near the paratope could have been modified). This is by no means to say that the conjugate is not fit-for-purpose, of course, as seen by the subsequent functional data.

Line 253 – Any comments why the level of mPD-1 seems so variable in BF16F10 cells over time while actin is constant? Fig S4/H

Line 274 – Should be syngeneic mouse models, right?

Line 316 – I would use the term “off-cancer” rather than “off-tissue” for clarity (once could take “off-tissue” to apply more broadly to the tissue type that gave rise to the cancer)

Line 323 – Again, I would not use the term “significantly” without backing it up statistically (though I agree that this *would* be statistically significant). Perhaps use “notably” instead? Or run a t-test.

Line 331 – Again, I would not use the term “significantly” without backing it up statistically. Please, either run stats on the data in Fig S5, or use a different term. With the outlier in S5/C, the statistical significance is not immediately apparent. Again, I think “notable” (or a synonym) would be an appropriate term.

Line 338 – Any comment on why you didn’t use SiRNA against both FR1 and FR2 at the same time? I.e., to show whether the FRTACS only relied on these two FR isoforms and no other receptor. I’m not suggesting you do this, but it would be good to hear your reasoning for this. If you had good reasons, perhaps worth including that in the discussion here.

Line 342 – I would include that it seems to be the ratio of FR:target that is important. As I said before, that seems like a key finding.

Line 343 – What do you mean by “synergistic interactions”? I don’t recall this being discussed earlier or in relation to data. Either explain what leads you to this conclusion or remove this.

Line 352 – Again, I would not use the term “significantly” without backing it up statistically, and I see no stats in Fig 5/A and B. Just say “notably” etc.

Line 354 – I would not say “indicates the low toxicity” since you didn’t directly measure toxicity in Fig 5, but PD-L1 levels. You could say “indicates the potential for lower toxicity”.

In fact in the Fig 4 data you don’t see any toxicity associated with PD-L1 blockade from Ab3 (by body weight), thus it would be hard to say that Ab3-FA (also no toxicity) was any less toxic in this experiment. Perhaps at higher doses differences would become visible, and certainly having higher efficacy with no increase in toxicity is a good result in of itself. I agree that Ab3-FA *should* be less toxic, but you should not claim this until you have evidence.

Line 361 – I would use off-cancer, as before.

Line 368 – I would use off-cancer, as before.

Line 386 – I would use off-cancer, as before.

Line 548 – Should be syngeneic mouse models, right?

Reviewer #3:

Remarks to the Author:

In this manuscript entitled “Development of Folate Receptor Targeting Chimeras (FRTACs) for Cancer Selective Degradation of Extracellular”, Zhou and colleagues developed Folate Receptor Targeting Chimeras (FRTACs), which are folate-conjugated antibodies that could selectively degrade extracellular and transmembrane proteins. They demonstrated the degradation of extracellular proteins by an artificial system, and they then verified the degradation effect of several cancer-relevant membrane proteins including EGFR, PD-L1, and CD47 in cells. They further demonstrated the molecular mechanism which was dependent on the endosome-lysosome System. Finally, they evaluated the anti-tumor effect of Ab3-FA for the degradation of PD-L1 in a xenograft model. In general, the study is somewhat too similar to the studies of Lysosome Targeting Chimeras (LYTACs). The overall concept and design were the same, the engaged receptor is changed but LYTAC studies have already demonstrated the concept of engaging cell-type specific receptors such as ASPGR. Even the targets selected for degradation were almost the same as LYTAC studies. Thus, the overall novelty of the study seems to be very limited. In addition, there are several technical issues that need to be addressed to demonstrate the proposed degradation mechanisms:

1. Figure 1: The authors need to provide evidence that lysosome degradation is blocked by the CQ treatment in Fig. 1E. In addition, imaging assays need to be performed to validate the target's entering into the endosomes/lysosomes. Finally, genetic knockouts of key endocytosis genes such as Rab7 are required to further demonstrate the mechanism.
2. Figure 2: most of the data lacked quantifications of multiple replicates, which are required to calculate the DC50 and Dmax. The authors should also label the cell lines used in each panel.
3. Figure 2B: The degradation of EGFR by Ctx-FA did not exhibit a typical hook effect (Figure 2B), contradictory to the authors' statement in line 113. The authors need to test more concentrations and replicates to resolve the discrepancies.
4. Figure 2D: the images were not clear enough, and the FRTAC-treated group seems to have dimmer fluorescence and lower resolution rather than changes in localization. It's better to enlarge part of the images to illustrate the point and also perform quantifications of the distribution of fluorescence. In addition, FA conjugated with other antibodies or FA alone should be carried out as controls.
5. Figure 2E: There is also been a lack of control experiments, for example, Ctx alone/control FRTACs/FA alone. It is also better to perform live cell imaging to with lyso-trackers to demonstrate the internalization of EGFR to lysosomes.
6. Figure 2F: Bafilomycin had very limited effects, especially compared to the elevated baseline upon Baf treatment. This is concerning and the authors need to obtain clearer results, for example, by knocking out essential genes in the endosome-lysosome pathway.
7. An additional mechanistic question. As the authors have mentioned, “FR is recycled

back to the membrane for the transport of more folate-conjugates". For transmembrane targets, do they dissociate from FR after endocytosis? If not, they will be recycled back to the membrane with FR. If yes, FRTACs then only enhanced their endocytosis without influencing their sorting to the lysosomes because they dissociate FRs. In this case, did FRTACs just trigger the endocytosis of the target or did they also change the targets' sorting to the lysosomes? This needs to be further validated. This is especially important for recycling receptors such as PD-L1.

8. Figure 4 : How did the authors determine the concentrations used in vivo?

Pharmacokinetic experiments showing the concentrations of Ab3-FA in blood and in tumor tissue are required.

9. Figure S5c : no statistics. The data from the Ab3-FA group was also too variable to demonstrate the effect.

10. Figure 5A-B: the authors need to show the blots detecting FR levels to see how big the difference was among the cell lines tested.

11. Figure 5C-D: the authors should provide the complete FACS results with the distribution of FR signals with controls validating the signal specificity (for example, FR knock-down or knockouts). The data presented here showed only very mild increases in FR levels in Fadu and Huh7 cells, and it is hard to imagine this difference may provide complete cell-type specificity proposed and detected by the authors.

12. Figure 5F: a few selected images are unconvincing to demonstrate the lack of changes in PD-L1 levels. Please perform western blots or FACS from multiple animals for more quantifiable results.

13. Many figures in the paper lacked quantifications from multiple biological replicates and statistical analysis (Figures 2, 5, and most of the supplementary figures).

August 20, 2024

Dear Editor,

We would like to thank all reviewers for their constructive comments and suggestions regarding our manuscript. We have carefully considered their feedback and revised the manuscript accordingly to address their concerns. In particular, we have conducted several new experiments, including in-vivo studies, to strengthen our findings. Detailed point-by-point responses and the changes made in the revised manuscript are listed below.

Reviewer #1

Targeted protein degradation (TPD) technology is an emerging concept in recent years. Since the core of the technology is to use the cell's own degradation mechanism to remove target proteins, in principle, there is almost no target limitation for TPD. However, TPD technology relies on the developer's rational use of the cell's own degradation pathways. Proteolysis Targeting Chimera (PROTAC) has been developed based on the E3 ubiquitin ligase-dominated degradation pathway. Lysosome Targeting Chimeras (LYTACs) have also been developed based on the cell's own lysosomal degradation pathway. In this manuscript, the authors creatively propose the use of folate receptor (FR)-mediated endocytosis for the degradation of cancer-associated extracellular and membrane proteins. Specifically, FR, a major biomarker for a variety of tumors, mediates the efficient endocytosis of folate and folate conjugates by cells. Taking advantage of this property, the authors used click chemistry to sequentially couple folate to antibodies to different proteins and characterized the degradation of FRTAC at extracellular soluble and endogenous membrane proteins. Based on this, the authors used PL-L1 as a target to reduce the protein amount of PD-L1 in mice by FR as a way to achieve anti-tumor effects. Thus, the use of this technology greatly expands the means by which TDP technology can degrade extracellular and membrane proteins, and is of great value in assisting in the treatment of tumors.

Response: We thank this reviewer for a very insightful summary of our work and appreciate the positive feedback.

The novel TPD system developed by the authors using the new technology pathway is certainly exciting. However several points should still be made to further improve this work.

Key points:

1. *Regarding the cellular imaging present in the manuscript (Fig.2d; Fig.2e), the authors should consider quantifying the amount of EGFR remaining as well as the co-localization of EGFR with lysosomes.*

Response: We repeated the experiment by treating cells with Ctx, free FA and the combination of Ctx and free FA, FA-labeled human IgG isotype and Ctx-FA for 24 h, and co-staining the cells with EGFR and LAMP1 antibodies to indicate the location of these two proteins. We quantified the fluorescence intensity of EGFR and analyzed the colocalization of internalized EGFR with lysosome by Pearson's correlation coefficients. Our results showed that EGFR was significantly reduced and colocalized with lysosome when treated with Ctx-FA, whereas other controls didn't deplete and relocate EGFR on the cellular membrane. These findings have been incorporated into the manuscript and are presented in Fig.2d.

2. *Regarding the multiple experiments in the manuscript (Fig.2d-h; Fig.4J-k; Fig.5f), it is necessary for the authors to include FA controls in the experimental design to ensure that the results are not due to the presence of FA. Alternatively, the target Ctx could be replaced with a protein with no target sequence and coupled with FA as a control. This will further make the experimental conclusion more credible.*

Response: We added FA controls into the following experiments as suggested by the reviewer. We compared the EGFR levels in cells treated with Ctx, free FA, the combination of Ctx plus free FA, FA-labeled human IgG isotype and Ctx-FA for 24 h by both immunofluorescent staining and western blot. Our results demonstrated that only Ctx-FA could transport EGFR into lysosome for degradation. The results have been added into the manuscript and presented in Fig.2d and supplementary Fig.3c. For the *in vivo* anti-tumor effect study, we treated CT26 tumor-bearing mice with PBS, IgG-FA, Ab3, and Ab3 + free FA respectively via IP injection every other day for a total of 5 doses and monitored the tumor size until day 20. We found that the non-targeting FA-conjugated IgG was not able to reduce the tumor growth compared to PBS-treated group. Mice treated with Ab3 or Ab3 + free FA exhibited similar trends in tumor growth. These findings were added into the manuscript and presented in supplementary Fig.6. For *in vivo* mechanism and cancer selectivity study, CT26 tumor-bearing mice were administered with PBS, IgG-FA, Ab3, and Ab3 + free FA respectively via IP injection continuously for three days. We then isolated the lung, spleen and tumor from each group and analyzed mouse PD-L1 expression level in those tissues by IHC and western blot. Our results indicated that the level of PD-L1 was comparable in spleen and lung among all the groups. Tumorous PD-L1 was significantly degraded in Ab3-FA treated mice, while other controls had minimal effect on downregulating tumorous PD-L1. Moreover, we also

analyzed the level of CD8 in tumor tissue and found that the mice treated with Ab3-FA significantly increased the level of CD8 in tumor compared to other controls. The results were added into the manuscript and presented in Fig.5e and supplementary Fig.7, 8c,d. In summary, all the data demonstrate that the presence of FA has no effect on targeted protein degradation and tumor growth.

3. *Regarding Fig.3F, the authors suggest that the addition of BAF can effectively inhibit Ctx-FA-mediated EGFR degradation. However, it seems that the inhibition effect of BAF is significantly less effective compared to (Fig.S4I; Fig.4j) or the FA over-addition in Fig.3F. The authors need to further discuss the presentation of this result with a revised description of the conclusions. Are there other downstream pathways that dominate FR-mediated degradation.*

Response: We agree with the reviewer that the rescue of the EGFR degradation by BAF is moderate compared to the abolishment of mouse PD-L1 degradation and soluble protein degradation. The response to BAF might be cell line- and target protein-dependent. We have modified our description of this finding in the manuscript “the degradation of EGFR was partially abolished when the cells were treated with lysosomal degradation inhibitor, Bafilomycin A1”. To further confirm that the degradation mainly occurred in lysosome, we knocked down Rab7, the essential gene in the endosome/lysosome pathway, and treated cells with Ctx-FA. Our results showed that the downregulation of Rab7 could also partially abolish EGFR degradation in the presence of Ctx-FA, suggesting lysosome is involved in FRTAC-induced protein degradation. Additionally, we also demonstrated the colocalization of EGFR and lysosomes after Ctx-FA treatment by immunofluorescent staining, which provided further evidence that FRTAC could transport membrane protein into lysosome for degradation. To investigate whether FRTAC could induce the protein degradation through ubiquitin-proteasome degradation pathway, which is another major pathway for protein depletion, we co-treated cells with Ctx-FA and proteasome inhibitor MG132 and found that EGFR degradation induced by Ctx-FA could not be inhibited. This result suggests that the proteasome is not involved in the FRTAC-mediated protein degradation. We acknowledge that other pathways may exist, and while they are beyond the scope of this work, we will continue to explore them in future studies.

4. *Regarding the presentation of the results of Fig.4J-K, the authors performed mouse tumor formation experiments with B16F10, CT26, MOC1, but only the results of CT26 and MOC1 were presented. the presentation of the results of B16F10 is necessary. Also as mentioned in #2, the experiment needs to add further reasonable controls.*

Response: We attempted to isolate B16F10 tumors from mice at the end of the treatment, but the tumor was too soft, which makes it difficult to obtain intact tumors from the mice for analysis. Following the reviewer's suggestion, we included additional controls, IgG-FA and Ab3+free FA, for the *in vivo* anti-tumor effect study. CT26 tumor-bearing mice were treated with PBS, IgG-FA, Ab3, and Ab3 + free FA respectively via IP injection every other day for a total of 5 doses. We monitored the tumor growth until day 20 and demonstrated that FA had no impact on tumor growth. These results has been added into the manuscript and presented in supplementary Fig.6.

5. *Regarding the demonstration of the tumor inhibition effect of Ab3-FA in Fig.4, it seems that the tumor inhibition effect of Ab3-FA is not greatly enhanced compared to the Ab3 use group. Is it possible to add more positive control and negative reference to this experiment to better evaluate the effect of FRTACs.*

Response: We acknowledge that the tumor inhibition effect of Ab3-FA is not dramatically enhanced compared to Ab3 in all three models. This observation aligns with our understanding of the mechanisms at play, where Ab3 also facilitates the infiltration of CD8a+ cells into the tumor and potentially recruits other immune cells, contributing to its anti-tumor effects through Antibody-Dependent Cellular Cytotoxicity (ADCC) and Complement-Dependent Cytotoxicity (CDC). In contrast, Ab3-FA is internalized into the cell and degraded along with its targets. However, the cancer-targeting ability of the folate ligand, combined with the degradation mechanism, can counterbalance this effect. It is noteworthy that Ab3-FA demonstrated statistically significant improvement in tumor growth inhibition compared to Ab3 in all three models. We have added the relevant controls, as discussed earlier, to ensure the observed effects are specifically related to FRTAC.

6. *The authors state in Fig.S5a-b that the effect of Ab3-FA was a decrease in PD-L1 expression as well as an increase in the infiltration of CD8a+ cells in the tissue. Whether the use of antibodies to CD8a in this use could antagonize the tumor inhibitory effect of Ab3-FA. Also compared to the results of Fig.4J-K, the rise in the number of CD8a+ cells is significant, how to understand the results of this experiment that the tumor inhibitory effect is not strong.*

Response: It has been shown that CD8 antibody could reverse the anti-tumor activity of PD-1/PD-L1 blockade. Examples are listed below.

Nat Commun 9, 4586 (2018). <https://doi.org/10.1038/s41467-018-06890-y>*Oncoimmunology* 5, e1238557 (2016). <https://doi.org/10.1080/2162402X.2016.1238557>*Onco Targets Ther* 12, 6961-

6971 (2019). <https://doi.org/10.2147/OTT.S202941> *Oncoimmunology* **10**, 1956142 (2021).
<https://doi.org/10.1080/2162402X.2021.1956142>

CD8 antibody should also inhibit the anti-tumor effect of our FRTAC against PD-L1. We plan to explore this in detail in future studies of our degrader. We observed that compared to PBS-treated mice, the treatment of Ab3-FA significantly delayed the tumor growth in all three mouse models. This finding is consistent with the significant increase of CD8a+ cells in Ab3-FA-treated mice. The difference between Ab3-treated group and Ab3-FA treated group is not dramatic in all three models, likely due to additional tumor killing effects such as ADCC or CDC induced by the antibody as discussed before. However, overall, Ab3-FA showed a statistically significant better tumor growth inhibition effect than Ab3. We believe that the cancer-targeting ability of the folate ligand, combined with the degradation mechanism, are the two main contributors to the observed difference in anti-tumor effects.

7. *Besides PD-L1, what are the other applicable targets for FRTACs? The authors have a powerful system for degradation of extracellular and membrane proteins, can they find more suitable targets for this technology to achieve better inhibition of tumor progression?*

Response: We have added a discussion of potential targets of FRTAC in the manuscript as suggested by the reviewer. “Beyond the targets examined in this paper, FRTAC can also be applied to various other cancer-relevant targets. These include matrix metalloproteinases and heparanase, important for tumor invasion and metastasis, as well as pH regulator carbonic anhydrase IX, which is related to tumor metabolism and microenvironment. The cancer selectivity of FRTAC has the potential to reduce the on-target off-cancer toxicity associated with current therapeutic inhibitors of these targets due to low expression level in normal tissues.”

8. *The authors have developed a set of protein degradation tools based on endocytosis of cell surface proteins. In addition to the excellent endocytosis receptor FR of tumor cells, the authors should discuss whether other endocytosis receptor pathways can be used to expand the use of this technology.*

Response: As suggested, we have expanded the discussion to include other cancer-overexpressing receptors that have been reported for lysosomal targeting degradation and proposed three receptors with distinct expression locations that can be used for tissue or cell-specific targeting.

“Although integrin and transferrin receptor have been employed as LTRs, no cancer selectivity was demonstrated. Other recycling receptors with selective

expression pattern may contribute to the tissue specific degraders, such as follicle-stimulating hormone receptor and prostate-specific membrane antigen.”

Reviewer #2

The authors of this manuscript report on the development of Folate Receptor Targeting Chimeras (FRTACs) which are a new modality for extracellular targeted protein degradation (ETPD) which is an up-and-coming area of research. The novelty of the work lies in the new degradation-inducing ligand (folic acid, FA) and the concept of FRTACs being selective for cancer. While there is another recent example of a cancer-selective ETPD moiety (which targets integrins, Zheng et al., JACS, 2022) that work did not examine the selectivity of their reagent for cancer vs healthy tissues. Also important to note is that FA has been used as a cancer-targeting agent for PROTACs (referenced in the manuscript) in a strategy that relied on FA-mediated internalization, but this application is certainly different enough to merit novelty.

The work reported in this manuscript is of high quality and rigor. Appropriate controls are included and FRTACs are examined from multiple different angles. The results are promising, with a PD-L1-FRTAC showing increased in vivo efficacy compared to PD-L1 blockade by the parent antibody at the same concentration. PD-L1-FRTAC also showed lower degradation of PD-L1 in healthy tissue than in cancer cells, suggesting it is indeed selective, which could very well translate to higher safety –although this wasn’t conclusively demonstrated here. I think the data showing that the ratio of target:lysosomal targeting receptor (LTR) is a key determinant of ETPD efficacy is a key finding as well, which could bear more highlighting in the manuscript. Perhaps this is generalizable to other ETPD systems and would be useful to bear in mind for others working in the field. The manuscript is well-written and easy to follow in general, though it would benefit from some additional proofreading. I have provided line-by-line points below, but these are relatively minor issues.

I do, however, have some issues with the MS data. Most of the MS spectra are up to standard, but in some of them the signal-to-noise ratio is so low, I do not think meaningful conclusions can be drawn.

*These are: *S7a/Ab-FA*, *S7b/NHS_12x (PEG2k)*, *S7d/Atz-FA*, S7e/Ab2-FA, S7f/Ab3 (a native antibody should look better!), *S7f/Ab3-FA-12x*, *S7f/Ab3-25x*. I have starred the worst offenders, and these need to be repeated. I think repeating the two unstarred ones would also improve the manuscript, but these are just about acceptable as they are.*

Your MALDI data in your 2021 ACS Central Science paper are a good example of good MALDI spectra as well as those from this manuscript I have not highlighted. I do understand that it is hard to get clear MS data for heterogeneous NHS-ester modified conjugates, but if you are unable to get meaningful data this way, a different form of

analysis is required. Some form of HPLC perhaps? SEC or HIC? These would give less precise data, but at least you could get better signal-to-noise and have a better chance of actually interpreting it.

In summary, I would recommend accepting this manuscript provided the above MALDI data are improved, or additional analysis is used to characterize the highlighted conjugates in a satisfactory manner, and if my below comments are addressed.

Response: We thank the reviewer for the positive assessment of our manuscript and appreciate the suggestion for further improving our manuscript. We repeated the MALDI mass analysis of the samples listed above with a modified method to increase signal-to-noise ratio. The modified method is described in the method section and the new high-quality data are presented in supplementary Fig 10. The detailed responses to the line-by-line comments are provided below.

Specific line-by-line comments:

Line 11 – target through the cell’s own disposal systems

Response: we made the change as suggested by the reviewer.

Line 15 – which would

Response: we made the change as suggested by the reviewer.

Line 20 – Look into formatting of in vivo and in vitro

Response: we made all “in vivo” and “in vitro” italic.

Line 29 – PROteolysis TArgeting Chimeras (PROTACs) were developed

Response: we made the change as suggested by the reviewer.

Line 30 - Over a dozen of PROTACs that

Response: we changed “a dozen of” to “a dozen”.

Line 32 - In addition, about a dozen of other

Response: we changed “a dozen of” to “a dozen”.

Line 33 - demonstrated in their utility

Some language editing would be needed (will not highlight from now on). N.B.: The rest of the manuscript has a lower density of errors, but some additional proofreading would be beneficial.

Response: we made the changes as suggested by the reviewer. We carefully proofread the rest of the manuscript and corrected all the errors.

Line 35 – I think you should talk about the FR-targeted PROTACs a little, as that is relevant and, in some ways, similar to this work (but certainly different enough!)

Response: We included a description of FR-targeted molecular glue and PROTAC in the manuscript as suggested by the reviewer. “Recently, FR-targeting strategy has been explored in the field of TPD. Studies have demonstrated that attaching folate to molecular glues or PROTACs allows for the selective degradation of cytosolic proteins in cancer cells”.

Line 52 – Integrins are overexpressed on various cancers, thus the integrin-based LYTACs in your Ref 27 (Zheng et al. 2022) do represent selective degradation in a disease setting. I think this is a key reference that you should dwell on in more detail. If your work has advantages over this, either by virtue of folate-targeting or through experimental design, you can detail that here. I would highlight that your work actually examined the selectivity of the ETPD strategy for cancer vs. healthy tissue, which the integrin-paper did not.

Response: As suggested, we discussed more about the integrin-based LYTAC and emphasized the advantage of our FRTAC over the integrin-based LYTAC in the introduction and discussion section as shown below. We mentioned transferrin receptor 1 based LYTAC as well.

“Integrin and transferrin receptor, overexpressed in cancer cells, have been recently leveraged for TPD. Degraders that recruit these two receptors hold the possibility of selectively depleting the extracellular proteins in cancer cells. However, the cancer selectivity has not been examined yet.”

“More importantly, in addition to the overexpression of FR in various cancers, the inaccessibility of FR in normal tissues to the circulating FA conjugates, a characteristic not shared by other cancer-overexpressing receptors, enables selective degradation of cancer-relevant protein targets in cancer cells.”

“Recently, an integrin-facilitated lysosomal degradation strategy has been shown to be effective in depleting soluble and membrane targets by recruiting integrin upregulated on cancer cells. Transferrin receptor targeting chimeras has been developed to utilize transferrin receptor 1, which is overexpressed on cancer cells, for lysosomal targeting degradation. However, both systems still need to demonstrate their cancer selectivity.”

Line 145 – I would show the data from Fig S1D in Fig 1 as well. They complement each other well and it is good to have a side-by-side comparison

Response: We followed the reviewer’s comment and have moved the data from supplementary F1d to Fig 1k.

Line 168 – Any comment on why the 2-step protocol worked better? I cannot think of any obvious explanation if the same degree of labelling is achieved in both cases

Response: We repeated the experiment and conducted statistical analysis. The results showed most degraders with the same linker length and degree of labeling showed no significant difference in EGFR degradation efficiency between the one-step and two-step labeling methods, the degrader with a 2k-PEG-linker which was synthesized using the two-step labeling method (N3, 2k, 25x) showed higher degradation efficiency than the degrader with either 1k or 2k PEG-linker, which were generated by one-step labeling method. Thus, we used this labeling method for the following experiments. We have corrected our description in the manuscript and presented the data with statistical analysis in supplementary Fig2.

Line 172 – Any comment on why 2k-PEG was better than 1k-PEG? Longer length allowing dual engagement of both receptors better?

Response: We repeated the experiment and conducted statistical analysis. The results showed that there was no significant difference in the EGFR degradation efficiency mediated by the degrader with either 1k- or 2k-PEG-linker when the degraders had similar degree of labeling and generated by the same labeling method. The 1k PEG appears to be long enough for the dual engagement of both receptors. We have corrected our description in the manuscript and presented the data with statistical analysis in supplementary Fig2.

Line 750 – Cell type (Fadu, right?) not noted in Fig caption for B-G

Response: We have included the cell type into each figure and figure caption.

Line 206 – It's an interesting concept that the ratio of EGFR:FR is what really matters, and it is great that you have established this. You could explain this in more detail in the text, as I think this is a key point for ETPD in general!

Response: We described more about the results related to the ratio of FR to EGFR in the manuscript and explained more in discussion.

“The degradation efficiency of Ctx-FA correlated more strongly with the ratio of FR to EGFR expression levels than with FR expression level alone, due to the involvement of both proteins in the ternary complex. A higher FR to EGFR expression ratio resulted in increased EGFR degradation efficiency (Supplementary Fig.3e-j).”

Line 246 – I would not use the term “significantly” without backing it up statistically. I would also argue that a 3-fold decrease is notable, and could be attributable to the random nature of lysine-conjugation (i.e., lysines near the paratope could have been

modified). This is by no means to say that the conjugate is not fit-for-purpose, of course, as seen by the subsequent functional data.

Response: We repeated the MST assay and determined the Kd of Ab as 9.05 ± 0.78 nM and Ab3-FA as 24.6 ± 4.67 nM. We conducted statistical analysis using an unpaired two-tail t-test and the results showed that there was no significant difference in the Kd between Ab3 and Ab3-FA. We have included the figure with statistical analysis in supplementary Fig.4b.

Line 253 – Any comments why the level of mPD-1 seems so variable in BF16F10 cells over time while actin is constant? Fig S4/H

Response: We repeated the experiment by inducing mouse PD-L1 expression for the same time to ensure the basal level of mPD-L1 is similar among different time points. We replaced the figure with statistical analysis in supplementary Fig.4e,f.

Line 274 – Should be syngeneic mouse models, right?

Response: we changed to “syngeneic” mouse as suggested by the reviewer.

Line 316 – I would use the term “off-cancer” rather than “off-tissue” for clarity (once could take “off-tissue” to apply more broadly to the tissue type that gave rise to the cancer)

Response: we changed to “off-cancer” as suggested by the reviewer.

*Line 323 – Again, I would not use the term “significantly” without backing it up statistically (though I agree that this *would* be statistically significant). Perhaps use “notably” instead? Or run a t-test.*

Response: We repeated the experiments with new cancer cell lines to compare EGFR and PD-L1 degradation in normal and cancer cells. We also characterized FR expression level in these cells and did statistical analysis for these data. We have added the description in the manuscript and replaced the figure with statistical analysis in Fig.5a-c.

Line 331 – Again, I would not use the term “significantly” without backing it up statistically. Please, either run stats on the data in Fig S5, or use a different term. With the outlier in S5/C, the statistical significance is not immediately apparent. Again, I think “notable” (or a synonym) would be an appropriate term.

Response: We repeated the *in vivo* mechanism in CT26 tumor mice. CT26 tumor-bearing mice were administered with PBS, IgG-FA, Ab3, and Ab3 + free FA respectively via IP injection continuously for three days. We then isolated the tumor from each group and analyzed mouse PD-L1 and CD8 expression level in tumors by IHC and western

blot. We quantified our results from western blot and did the statistical analysis. Our results demonstrated that tumorous PD-L1 was significantly degraded in Ab3-FA treated mice, while other controls had minimal effect on downregulating tumorous PD-L1. Moreover, mice treated with Ab3-FA showed significantly increased the level of CD8 in tumor compared to other controls. The results were added into the manuscript and presented in supplementary Fig.7.

Line 338 – Any comment on why you didn't use SiRNA against both FR1 and FR2 at the same time? I.e., to show whether the FRTACS only relied on these two FR isoforms and no other receptor. I'm not suggesting you do this, but it would be good to hear your reasoning for this. If you had good reasons, perhaps worth including that in the discussion here.

Response: We attempted to use siRNA to knock down FR2 but were unsuccessful. We also tried other methods, such as CRISPER, shRNA, to generate FR2 knockdown/knockout cell line, but these efforts have not been successful to date.

Line 342 – I would include that it seems to be the ratio of FR:target that is important. As I said before, that seems like a key finding.

Response: As suggested, we emphasized the importance of FR:target expression ratio in the discussion.

“Degraders exhibit lower degradation efficiency when cells express abundant amounts of target protein but low levels of FR. This is likely due to insufficient FR-mediated targeting, leading to suboptimal engagement of the degraders with the target protein. The ratio between FR and endogenous target expression levels is a better determinant of membrane protein degradation compared to FR expression alone.”

Line 343 – What do you mean by “synergistic interactions”? I don't recall this being discussed earlier or in relation to data. Either explain what leads you to this conclusion or remove this.

Response: We removed the “synergistic interactions” as suggested by the reviewer.

Line 352 – Again, I would not use the term “significantly” without backing it up statistically, and I see no stats in Fig 5/A and B. Just say “notably” etc.

Response: We repeated the experiments with new cancer cell lines and did statistical analysis for these data. We have added the description in the manuscript and replaced the figure with statistical analysis in Fig.5a-c.

Line 354 – I would not say “indicates the low toxicity” since you didn't directly measure toxicity in Fig 5, but PD-L1 levels. You could say “indicates the potential for lower

toxicity”.

*In fact in the Fig 4 data you don't see any toxicity associated with PD-L1 blockade from Ab3 (by body weight), thus it would be hard to say that Ab3-FA (also no toxicity) was any less toxic in this experiment. Perhaps at higher doses differences would become visible, and certainly having higher efficacy with no increase in toxicity is a good result in of itself. I agree that Ab3-FA *should* be less toxic, but you should not claim this until you have evidence.*

Response: we changed to “indicates the potential for lower toxicity” as suggested by the reviewer.

Line 361 – I would use off-cancer, as before.

Response: We indicate not only cancers but a broad range of pathogenic tissues, so we used “on target off tissue toxicity” here. We proposed three receptors with distinct expression locations that can be used for tissue or cell-specific targeting later.

Line 368 – I would use off-cancer, as before.

Response: we changed to “off-cancer” as suggested by the reviewer.

Line 386 – I would use off-cancer, as before.

Response: we changed to “off-cancer” as suggested by the reviewer.

Line 548 – Should be syngeneic mouse models, right?

Response: we changed to “syngeneic” mouse models as suggested by the reviewer.

Reviewer #3

In this manuscript entitled “Development of Folate Receptor Targeting Chimeras (FRTACs) for Cancer Selective Degradation of Extracellular”, Zhou and colleagues developed Folate Receptor Targeting Chimeras (FRTACs), which are folate-conjugated antibodies that could selectively degrade extracellular and transmembrane proteins. They demonstrated the degradation of extracellular proteins by an artificial system, and they then verified the degradation effect of several cancer-relevant membrane proteins including EGFR, PD-L1, and CD47 in cells. They further demonstrated the molecular mechanism which was dependent on the endosome-lysosome System. Finally, they evaluated the anti-tumor effect of Ab3-FA for the degradation of PD-L1 in a xenograft model. In general, the study is somewhat too similar to the studies of Lysosome Targeting Chimeras (LYTACs). The overall concept and design were the same, the

engaged receptor is changed but LYTAC studies have already demonstrated the concept of engaging cell-type specific receptors such as ASPGR. Even the targets selected for degradation were almost the same as LYTAC studies. Thus, the overall novelty of the study seems to be very limited.

Response: We appreciate the feedback and would like to address the concerns regarding the novelty and significance of our study in comparison to those from literature. We believe our work offers several distinct contributions to the field, which we detail below:

1. Disease-relevant Receptor Targeting Strategy: Degraders that can recruit LTRs primarily expressed in pathogenic tissues would significantly reduce potential toxicity caused by on-target off-tissue effects. In this aspect, the cancer-targeting FRTAC is unique compared to other reported lysosome-targeting degraders, which do not show selective degradation in disease-relevant tissues. Furthermore, the folate receptor (FR) is overexpressed in a broad range of cancer cells and has differential accessibility in normal tissues and tumors due to the difference between the apical surface of polarized epithelial cells in healthy tissues and tumor cells. This makes FR an ideal candidate for cancer-selective LTR. Our study uniquely demonstrates both in vitro and in vivo cancer selectivity, a feature not evidenced in the field of extracellular TPD.

2. Molecular Mechanism and Selectivity: Previous studies on lysosomal targeting degraders utilized LTRs for cell internalization via clathrin-mediated endocytosis; in contrast, the folate receptor employs caveolae-mediated endocytosis. We provided additional experimental evidence (Figure 1I) to support this mechanism. Our work opens a new avenue for exploring receptors with a different endocytosis mechanism. We also delved into the molecular mechanism of FRTACs, demonstrating their dependence on the endosome-lysosome system. We provided detailed evidence of how FRTACs leverage the FR to internalize and degrade extracellular and membrane proteins selectively in cancer cells. Additionally, we observed that a high concentration of free folic acid is necessary to counteract the degradation effect of FRTACs, addressing limitations seen with other LYTACs where effectiveness can be abolished by low concentrations of endogenous ligands.

3. Significance and Scope of Target Proteins: While some target proteins such as EGFR, PD-L1, and CD47 have been studied previously, our work is the first to show their degradation via FR-mediated targeting. This distinction is crucial as it expands the applicability of targeted degradation strategies to a broader range of cancers that overexpress FR, offering a new avenue for therapeutic intervention.

4. In Vivo Efficacy: Initial LYTACs that recruit CIM6PR and more recent LYTACs that recruit cell-type selective ASGPR only showed in-vivo degradation without efficacy

studies. We have demonstrated the anti-tumor effect of PD-L1 degraders in three different syngeneic mouse models by recruiting FR. This in vivo validation provides a deeper understanding of the mechanism of action and cancer selectivity. It highlights the therapeutic potential of FRTACs and provides a foundation for future clinical applications. The success of FRTAC in reducing tumor growth differentiates our work from most prior studies focused primarily on in vitro systems without in-vivo efficacy studies. The in-vivo anti-tumor effect was only reported using an integrin-facilitated lysosomal degradation strategy within the field of extracellular TPD. However, no cancer-selective degradation was demonstrated in either in-vitro or in-vivo experiments.

5. Comprehensive Analysis: Our study provides a comprehensive analysis of the degradation mechanism and scope of FRTACs on both extracellular and transmembrane proteins, showcasing the versatility and broad applicability of this approach. This thorough investigation adds depth to the current understanding of extracellular TPD.

In summary, while our study builds upon the foundational concepts introduced by LYTACs, including one that recruits ASGPR from our own lab, it presents significant advancements in terms of disease-relevant lysosome targeting receptors, mechanistic insights, and therapeutic potential. We believe these contributions offer substantial novelty and value to the field. We hope this clarifies the unique aspects of our work and addresses the concerns. Given the advancements and novel findings, we believe our work presents sufficient originality to warrant publication in Nature Communications.

In addition, there are several technical issues that need to be addressed to demonstrate the proposed degradation mechanisms:

1. *Figure 1: The authors need to provide evidence that lysosome degradation is blocked by the CQ treatment in Fig. 1E. In addition, imaging assays need to be performed to validate the target's entering into the endosomes/lysosomes. Finally, genetic knockouts of key endocytosis genes such as Rab7 are required to further demonstrate the mechanism.*

Response: It is generally believed that chloroquine functions as a lysosome degradation inhibitor by increasing the pH within lysosomes. This increase in pH disrupts the activity of lysosomal enzymes, which rely on an acidic environment to function effectively. Consequently, this disruption inhibits the lysosome's ability to degrade its contents. In addition to chloroquine, we also included Bafilomycin A1 (BAF), another commonly used lysosome degradation inhibitor, and proteasome degradation inhibitor MG132. Our results showed that both chloroquine and BAF could significantly inhibit the degradation, while MG132 had no effect on rescuing the degradation. These data demonstrate that lysosome but not proteasome is

involved in FRTAC-induced protein degradation. These results have been included into the manuscript and the data were presented in Fig.1e.

As suggested, we performed the live cell imaging of cells treated with Ab-FA and anti-Rabbit-647 to monitor the trafficking of anti-Rabbit-647 after internalization. Cells were also treated with LysoTracker, which indicates the location of lysosome. Cells treated with Ab and the combination of Ab plus free FA were used as negative controls. We quantified the intracellular fluorescence intensity and analyzed the colocalization of anti-Rabbit-647 with lysosomes. Our results indicated that our Ab-FA could significantly facilitate the uptake of anti-Rabbit-647 compared to Ab and Ab + free FA. The internalized anti-Rabbit-647 was colocalized with lysosomes, confirming that the target protein was delivered into lysosomes after internalization. These results have been added into the manuscript and are presented in Fig.1f.

Lastly, we knocked down Rab7 in Hela cells and found that the amount of intracellular anti-Rabbit-647 was significantly higher in the Rab7 knockdown cells compared to cells transfected with scramble siRNA. This suggests that the internalized anti-Rabbit-647 was accumulated in the cell with less degradation due to the disruption of endosome/lysosome pathway by downregulating Rab7 in the cells. These results have been added into the manuscript and are presented in Fig.1g. Overall, all the data support that FRTAC could transport the target protein into lysosomes for degradation.

2. *Figure 2: most of the data lacked quantifications of multiple replicates, which are required to calculate the DC50 and Dmax. The authors should also label the cell lines used in each panel.*

Response: We followed the reviewer's suggestion to obtain sufficient repeats for the experiments in the manuscript and conducted statistical analysis. We calculated the DC50 and Dmax for EGFR degradation in Fadu and Hela cells, as well as for mPD-L1 degradation in CT26 and B16F10 cells. These results have been added to the manuscript and are presented in Fig.2b, supplementary Fig.3a, and supplementary Fig.4c,d. In addition, we labeled the cell lines used in each panel as suggested by the reviewer.

3. *Figure 2B: The degradation of EGFR by Ctx-FA did not exhibit a typical hook effect (Figure 2B), contradictory to the authors' statement in line 113. The authors need to test more concentrations and replicates to resolve the discrepancies.*

Response: In line 113, the discussion on hook effect is associated with small molecule degraders. In the manuscript, we described that the uptake of anti-FITC-594 mediated by small molecule FA-FITC peaked at 200 nM and decreased at 1000 nM, reflecting a typical hook effect of the small molecule degrader. There is no discrepancy.

For the antibody-based degraders, including Ab-FA, Ctx-FA and Ab3-FA, they exhibited dose dependency without hook effect at the concentrations we tested. We typically treated cells with antibody-based degraders with up to 100 nM concentration because of their high potency.

4. *Figure 2D: the images were not clear enough, and the FRTAC-treated group seems to have dimmer fluorescence and lower resolution rather than changes in localization. It's better to enlarge part of the images to illustrate the point and also perform quantifications of the distribution of fluorescence. In addition, FA conjugated with other antibodies or FA alone should be carried out as controls.*

Response: As suggested, we enlarged part of the images, quantified the fluorescent intensity, and included additional controls. We repeated the experiment by treating cells with Ctx, free FA, the combination of Ctx and free FA, FA-labeled human IgG isotype, and Ctx-FA for 24 h. We then co-stained the cells with EGFR and LAMP1 antibodies to indicate the location of these two proteins. As recommended by the reviewer, we quantified the fluorescence intensity of EGFR and analyzed the colocalization of internalized EGFR with lysosome using Pearson's correlation coefficients. Our results showed that EGFR was significantly reduced and colocalized with lysosome when treated with Ctx-FA, whereas the other controls didn't deplete and relocate EGFR on the cellular membrane. These findings have been incorporated into the manuscript and are presented in Fig.2d.

5. *Figure 2E: There is also been a lack of control experiments, for example, Ctx alone/control FRTACs/FA alone. It is also better to perform live cell imaging to with lyso-trackers to demonstrate the internalization of EGFR to lysosomes.*

Response: As suggested by the reviewer, we included additional controls: Ctx, free FA, the combination of Ctx and free FA, and FA-labelled human IgG isotype. Staining EGFR with a detecting antibody in live cells is challenging. Instead, we co-stained the cells with EGFR and LAMP1 antibodies to indicate the location of these two proteins. We then analyzed the colocalization of internalized EGFR with lysosome using Pearson's correlation coefficients. Our results showed that EGFR was significantly reduced and colocalized with lysosome when treated with Ctx-FA, whereas the other controls didn't deplete and relocate EGFR from the cellular

membrane. The results have been added to the manuscript and are presented in Fig.2d.

6. *Figure 2F: Bafilomycin had very limited effects, especially compared to the elevated baseline upon Baf treatment. This is concerning and the authors need to obtain clearer results, for example, by knocking out essential genes in the endosome-lysosome pathway.*

Response: We acknowledge the reviewer's concern that the rescue of the EGFR degradation by BAF is moderate, though statistically significant. However, the abolishment of mouse PD-L1 degradation and soluble protein degradation is much more significant, suggesting that the response to BAF might be cell line- and target protein-dependent. We have revised our description of this finding in the manuscript to state: "the degradation of EGFR was partially abolished when the cells were treated with lysosomal degradation inhibitor, Bafilomycin A1".

To further confirm that the degradation mainly occurred in lysosome, we knocked down Rab7, an essential gene in the endosome/lysosome pathway, and treated cells with Ctx-FA. Our results showed that downregulation of Rab7 could also partially abolish EGFR degradation in the presence of Ctx-FA, suggesting lysosome is involved in FRTAC-induced protein degradation.

Additionally, we demonstrated the colocalization of EGFR and lysosomes after Ctx-FA treatment using immunofluorescent staining, providing further evidence that FRTAC could transport membrane protein into lysosome for degradation. To investigate whether FRTAC could induce the protein degradation through ubiquitin-proteasome degradation pathway, another major pathway for protein degradation, we co-treated cells with Ctx-FA and proteasome inhibitor MG132. We found that EGFR degradation induced by Ctx-FA could not be inhibited by MG132. This result suggests that the proteasome is not involved in the FRTAC-mediated protein degradation.

7. *An additional mechanistic question. As the authors have mentioned, "FR is recycled back to the membrane for the transport of more folate-conjugates". For transmembrane targets, do they dissociate from FR after endocytosis? If not, they will be recycled back to the membrane with FR. If yes, FRTACs then only enhanced their endocytosis without influencing their sorting to the lysosomes because they dissociate FRs. In this case, did FRTACs just trigger the endocytosis of the target or did they also change the targets' sorting to the lysosomes? This needs to be further validated. This is especially important for recycling receptors such as PD-L1.*

Response: We observed that the transmembrane target EGFR was degraded while the level of FR remained unchanged in the presence of FRTACs. This suggests that EGFR dissociates from FR within the endosome before being transported to the lysosome for degradation. EGFR has been reported to traffic to recycling endosome, lysosome, mitochondria and nuclei after ligand or stress induced internalization, indicating that multiple trafficking pathways for EGFR could operate following internalization. (*Trends Cell Biol* **26**, 352-366 (2016). <https://doi.org/10.1016/j.tcb.2015.12.006>) Thus, FRTAC could not only trigger the endocytosis of EGFR but also specifically direct its trafficking to the lysosome rather than other compartments.

PD-L1 undergoes continuous endocytosis, with a large portion being recycled back to the cell membrane to maintain its surface levels. PD-L1 can also be degraded through either lysosome or proteasome degradation pathway, depending on different modifications and molecules involved. (*Cancer Immunol Res* **11**, 866-874 (2023). <https://doi.org/10.1158/2326-6066.CIR-22-0953>) We demonstrated that FRTAC could significantly reduce PD-L1 level through lysosomal degradation pathway. These results suggested that FRTAC could redirect a substantial portion of PD-L1 from its recycling route to the lysosomal degradation pathway.

Overall, our results indicate that FRTAC could both promote the endocytosis of the membrane target proteins and impact their subsequent sorting, specifically promoting their degradation via the lysosomal pathway.

We acknowledge that additional experiments could be conducted to further elucidate the details of the mechanism. While they are beyond the scope of this work, we will continue to explore them in future studies.

8. *Figure 4 : How did the authors determine the concentrations used in vivo? Pharmacokinetic experiments showing the concentrations of Ab3-FA in blood and in tumor tissue are required.*

Response: To determine the concentrations of Ab3 and Ab3-FA used in vivo, we initially used published studies as the reference. These studies typically used anti-mouse PD-L1 antibodies at a dosage of 10 mg/kg for in vivo experiments.

Examples of them are shown below.

Sci Rep **8**, 217 (2018). <https://doi.org/10.1038/s41598-017-18641-y>

J Clin Invest **128**, 1708 (2018). <https://doi.org/10.1172/JCI120803>

Nat Commun **8**, 14572 (2017). <https://doi.org/10.1038/ncomms14572>

Oncoimmunology 6, e1294299 (2017). <https://doi.org/10.1080/2162402X.2017.1294299>

Based on these reports, we conducted experiments to test various concentrations of Ab3 and Ab3-FA across three different mouse models to establish the appropriate dosing.

In response to the reviewer's suggestion, we performed pharmacokinetic experiments to assess the plasma concentration of Ab3-FA. We administered Ab3-FA at a dose of 2.5 mg/kg via intraperitoneal (IP) injection to B16F10 tumor-bearing C57BL/6 mice and collected plasma samples at different time points. The results of these experiments are included in the manuscript and presented in Figure 3b and Supplementary Figure 5b. Regarding the measurement of Ab3-FA concentration in tumor tissues, we encountered challenges. Although we attempted to detect Ab3-FA in tumor samples, the results were inconclusive, showing smears rather than distinct bands. This may be due to varying degree of degradation of the degrader along with PD-L1 in the lysosome, complicating the detection of Ab3-FA within the tumor tissue.

9. *Figure S5c : no statistics. The data from the Ab3-FA group was also too variable to demonstrate the effect.*

Response: We repeated the *in vivo* mechanistic studies in CT26 tumor-bearing mice. CT26 tumor-bearing mice were administered with PBS, IgG-FA, Ab3, and Ab3 + free FA, respectively, via IP injection once a day for three days. We then isolated the tumor from each group and analyzed mouse PD-L1 and CD8 expression level in tumors by IHC and western blot. We quantified the results from western blot and did the statistical analysis. Our results demonstrated that tumorous PD-L1 was significantly degraded in Ab3-FA treated mice, while other controls had minimal effect on downregulating tumorous PD-L1. Moreover, mice treated with Ab3-FA showed significantly increased the level of CD8 in tumor compared to other controls. The results were added into the manuscript and are presented in supplementary Fig.7.

10. *Figure 5A-B: the authors need to show the blots detecting FR levels to see how big the difference was among the cell lines tested.*

Response: In response to the reviewer's suggestion, we have now included Western blot analyses to assess the levels of folate receptors (FR) in the cell lines HACAT, Huh7, and TU138. The western blot data show that the difference in FR levels between TU138 and HACAT is approximately 5-fold, and the difference

between Huh7 and HACAT is 8-fold. The updated results have been added to the manuscript and are presented in Figure 5c.

11. *Figure 5C-D: the authors should provide the complete FACS results with the distribution of FR signals with controls validating the signal specificity (for example, FR knock-down or knockouts). The data presented here showed only very mild increases in FR levels in Fadu and Huh7 cells, and it is hard to imagine this difference may provide complete cell-type specificity proposed and detected by the authors.*

Response: To validate the specificity of the FR signal, we attempted to include controls using FR1/FR2 knockdown/knockout cells. However, despite numerous attempts over the past six months, we were unable to achieve double knockdown or knockout for FR1/FR2. We then quantified the FR level in difference cells by western blot analysis. The updated results have been added to the manuscript and are presented in Figure 5c. There is a significant difference between FR levels by Western blot, which is consistent with the selective degradation.

12. *Figure 5F: a few selected images are unconvincing to demonstrate the lack of changes in PD-L1 levels. Please perform western blots or FACS from multiple animals for more quantifiable results.*

Response: In response to the reviewer's request for more quantifiable results, we performed additional analyses to better demonstrate the lack of changes in PD-L1 levels in non-tumor tissues. For in vivo cancer selectivity study, CT26 tumor-bearing mice were administered with PBS, IgG-FA, Ab3, and Ab3 + free FA respectively via IP injection once a day for three days. We then isolated the lung and spleen in each group and analyzed mouse PD-L1 expression level in those tissues by IHC and western blot. Our results indicated that the level of PD-L1 was comparable in spleen and lung among all the groups. The results were added into the manuscript and are presented in Fig.5e and supplementary Fig.8c,d.

13. *Many figures in the paper lacked quantifications from multiple biological replicates and statistical analysis (Figures 2, 5, and most of the supplementary figures).*

Response: In response, we have revised the manuscript to include quantifications for all relevant data. Specifically, we have provided statistical analyses for multiple biological replicates of the data presented in Figures 2, 5, and the majority of the supplementary figures. The revised figures and corresponding statistical data are now included in the revised manuscript.

I believe we have addressed all concerns raised by the reviewers in the revised manuscript. Should you need any further information, please do not hesitate to contact me.

I look forward to hearing from you.

Sincerely,
Weiping Tang, Ph.D.
Janis Apinis Professor,
Vilas Distinguished Achievement Professor,
School of Pharmacy,
University of Wisconsin-Madison,
Madison, WI 53705-2222

Reviewers' Comments:

Reviewer #1

(Remarks to the Author)

The authors have addressed my previous concerns.

Reviewer #2

(Remarks to the Author)

Dear Authors,

Thank you for your efforts in improving the manuscript and for taking my comments and suggestions on board.

I recommend the article be accepted for publication.

Best wishes,

Peter Szijj

Reviewer #3

(Remarks to the Author)

While some of the initial concerns have not been sufficiently addressed, the authors have made a good effort to further validate the mechanism of FRTACs' effects. This reviewer thus agrees with the publication of the study.